# Enhanced Alkaline Hydrogen Evolution on $Gd_{1.0}/Nd_x$ (x = 0.5, 1.0, 3.0, and 6.0%)-Doped $TiO_2$ Bimetallic Electrocatalysts

Mohammed Alsawat [1,*], Naif Ahmed Alshehri [2], Abdallah A. Shaltout [3], Sameh I. Ahmed [4], Hanan M. O. Al-Malki [1], Manash R. Das [5,6], Rabah Boukherroub [7], Mohammed A. Amin [1,*] and Mohamed M. Ibrahim [1,*]

[1] Department of Chemistry, College of Science, Taif University, Taif 21944, Saudi Arabia; hh.1818@hotmail.com
[2] Physics Department, College of Science, Al-Baha University, Aqiq, Al-Baha 65431, Saudi Arabia; nalshehri@bu.edu.sa
[3] Spectroscopy Department, Physics Research Institute, National Research Centre, El Behooth St., Dokki, Cairo 12622, Egypt; aa.shaltout@nrc.sci.eg
[4] Department of Physics, College of Science, Taif University, Taif 21944, Saudi Arabia; sameh2977@gmail.com
[5] Advanced Materials Group, Materials Sciences and Technology Division, CSIR-North East Institute of Science and Technology, Jorhat 785006, India; mrdas@neist.res.in
[6] Academy of Scientific and Innovative Research (AcSIR), Ghaziabad 201002, India
[7] Centre National de la Recherche Scientifique, Université Lille, Université Polytechnique Hauts-de-France, UMR 8520, IEMN, F-59000 Lille, France; rabah.boukherroub@univ-lille.fr
[*] Correspondence: mosawat@tu.edu.sa (M.A.); mohamed@tu.edu.sa (M.A.A.); ibrahim@tu.edu.sa (M.M.I.)

**Abstract:** The work reports a facile synthesis of high thermally stable nanocrystalline anatase $TiO_2$ nanoparticles (NPs) doped with different atomic concentrations (0.5, 1.0, 3.0, and 6.0%) of $Gd^{3+}$ and $Nd^{3+}$ ions by a template-free and one-step solvothermal process, using titanium(IV) butoxide as a titanium precursor and dimethyl sulfoxide (DMSO) as a solvent. The structure and morphology of the $Gd^{3+}$, $Nd^{3+}$, and $0.5\%Gd^{3+}$-$0.5\%Nd^{3+}$/doped $TiO_2$ NPs have been characterized by using various analytical techniques. The $Gd^{3+}$/ and $Nd^{3+}$/$TiO_2$ molar ratios were found to have a pronounced impact on the crystalline structure, size, and morphology of $TiO_2$ NPs. X-ray diffraction (XRD) and X-ray photoelectron spectroscopy (XPS) studies revealed the proper substitution of $Ti^{4+}$ by $Gd^{3+}$ and $Nd^{3+}$ ions in the $TiO_2$ host lattice. The as-prepared $Gd_x/TiO_2$, $Nd_x/TiO_2$, and $Gd_{1.0}/Nd_x/TiO_2$ bimetallic NPs, x = 0.5, 1.0, 3.0, and 6%, have been investigated as electrocatalysts for hydrogen evolution reaction (HER) in 1.0 M KOH solution using a variety of electrochemical techniques. At any doping percentage, the $Gd_{1.0}/Nd_x/TiO_2$ bimetallic NPs showed higher HER catalytic performance than their corresponding counterparts, i.e., $Gd_x/TiO_2$ and $Nd_x/TiO_2$. Upon increasing the Nd content from 0.5 to 6.0%, the HER catalytic performance of the $Gd_{1.0}/Nd_x/TiO_2$ bimetallic NPs was generally enhanced. Among the studied materials, the bimetallic $Gd_{1.0}/Nd_{6.0}/TiO_2$ NPs emerged as the most promising catalyst with an onset potential of −22 mV vs. RHE, a Tafel slope of 109 mV $dec^{-1}$, and an exchange current density of 0.72 mA $cm^{-2}$. Such HER electrochemical kinetic parameters are close to those recorded by the commercial Pt/C (onset potential: −15 mV, Tafel slope: 106 mV $dec^{-1}$, and exchange current density: 0.80 mA $cm^{-2}$), and also comparable with those measured by the most active electrocatalysts reported in the literature. The synergistic interaction of Gd and Nd is thought to be the major cause of the bimetallic catalyst's activity.

**Keywords:** titanium dioxide; rare earth-doped $TiO_2$ nanoparticles; electrochemical; hydrogen generation; alkaline

## 1. Introduction

The ever-growing demand for clean, renewable, environmentally friendly, and cost-effective energy sources has resulted from meeting the needs of modern society, which is expanding and evolving quickly. Utilizing hydrogen ($H_2$) as a possible energy source with the possibility of replacing fossil fuels has proven to be a particularly alluring strategy in this context,

partly because of its high combustion heat (287 kJ/mol) and the release of green byproduct water [1]. A significant portion of $H_2$ is produced from fossil fuels using a steam reforming method that also produces $CO_2$, which is thought to be a potential cause of environmental global warming [2].

One of the most advantageous processes for producing very pure $H_2$ gas is alkaline water electrolysis. The effectiveness of alkaline water electrolysis is, however, constrained by the sluggish kinetics of the hydrogen evolution process (HER) in alkaline media [3–5]. Effective electrocatalysts are suggested for use as cathode materials in water electrolysis cells because they produce significant amounts of $H_2$ at low overpotentials by accelerating the kinetics of the HER. Electrode materials based on platinum (Pt) are the most reliable and effective HER electrocatalysts. The excessive cost of these components, however, raises the price of water electrolyzers. As a result, one of the primary goals for the efficient production of $H_2$ on a large scale is the deployment of innovative, affordable, and effective HER electrocatalysts [6,7].

For the purpose of achieving good electrochemical performances, researchers have designed nanostructures of various Pt group metals (Ru, Rh, Ir, and Pd) with high surface-to-volume ratios [8]. Nitrogen-doped reduced graphene oxide (rGO)-based Pt-$TiO_2$ nanostructures [9], monolayer Pd and Au supported on $Mo_2C$ [10], monolayer Pt supported on bulk tungsten carbide (WC) [11], etc., were employed as HER electrocatalysts, demonstrating outstanding activity.

Extensive research has been conducted on alternative non-noble (Fe, Co, Ni, Mo), $MoS_2$ coupled with a perovskite oxide, and metal-free (carbon-based) electrocatalysts in an effort to avoid using precious noble metals [12–17]. Transition metal chalcogenides [18], carbides [19], metal alloys [20], and complexes [21] are examples of other non-noble electrocatalysts. Bimetallic NPs, which are composed of two elements, have drawn a lot of attention for their efficiency as HER electrocatalysts as well as for their potential use in a variety of energy storage applications [22–25]. They have greater catalytic characteristics compared to their monometallic counterparts, which is primarily ascribed to their improved and tunable chemical, physical, and cooperative interactions [25,26]. Au-Pd [27,28] and Au-Ni [29] bimetallic electrocatalysts exhibited high catalytic performance for the HER among bimetallic alloys. Excellent HER activity may be found in alloy nanostructures made of Cu-Pt [30], Au-Pt [31], and other metals. Core-shell nanostructures can also take the place of Pt in a number of HER reactions. Examples of outstanding performances include Au@Pt [32], Cu@Pd/Ti [33], Au@Pd [34], Au@Pd [35], and Au@CdS core-shell nanostructures [36].

In this work, a facile synthesis of $Gd^{3+}$ and $Nd^{3+}$-doped $TiO_2$ NPs at different atomic concentrations (0.5, 1.0, 3.0, and 6.0%) of $Gd^{3+}$ and $Nd^{3+}$ ions, as well as the mixed $0.5\%Gd^{3+}$-$0.5\%Nd^{3+}$-doped $TiO_2$ NPs, using titanium(IV) butoxide as a titanium precursor and dimethyl sulfoxide (DMSO) as a solvent, is presented. The lack of usage of amphiphilic surfactants, capping agents, or block copolymers, in contrast to the majority of earlier preparation techniques, can significantly reduce production cost of the synthesized catalysts. Our process is highly straightforward, economical, and easily scalable, as it required only three basic ingredients. The reducing agent and stabilizing layer in this approach were both performed by the DMSO solvent, which is believed to adhere to the surface of the NPs, preventing their agglomeration.

The structure and morphology of the obtained doped $TiO_2$ NPs, namely, $Gd_x/TiO_2$, $Nd_x/TiO_2$, and $Gd_{1.0}/Nd_x/TiO_2$ bimetallic NPs, x = 0.5, 1.0, 3.0, and 6%, were characterized by advanced surfaces analysis techniques. Using linear sweep voltammetry, the newly synthesized electrode materials were assessed for the first time as HER electrocatalysts in 1.0 M KOH aqueous solution. Chronoamperometry and repetitive cyclic polarization measurements were used to evaluate the stability of the best performing electrocatalyst.

## 2. Results and Discussion

### 2.1. Characterization of $Gd_x/TiO_2$, $Nd_x/TiO_2$, and $Gd_{0.5}/Nd_{0.5}/TiO_2$ NPs

The formation of $Gd_x/TiO_2$, $Nd_x/TiO_2$, and $Gd_{0.5}/Nd_{0.5}/TiO_2$ NPs was characterized by Fourier-Transform Infrared Spectroscopy (FT-IR) and simultaneous Thermogravimetric Analysis-Differential Thermal Analysis (TGA-DTA) (Supporting information, Figures S1 and S2).

Energy dispersive X-ray fluorescence (EDXRF) was used to quantify the elemental contents of the $Gd_x/TiO_2$, $Nd_x/TiO_2$, and $Gd_{0.5}/Nd_{0.5}/TiO_2$ NPs. Before carrying out the EDXRF measurements, a series of $Gd_x/TiO_2$, $Nd_x/TiO_2$, and $Gd_{0.5}/Nd_{0.5}/TiO_2$ NPs were homogenously distributed inside the powder sample holder of 25 mm diameter. Mylar foil (4.5 μm-thick, Chemplex Industries, Inc., Palm City, FL, USA) was installed inside the sample holder. Furthermore, fine powders of $Gd_x/TiO_2$, $Nd_x/TiO_2$ and $Gd_{0.5}/Nd_{0.5}/TiO_2$ NPs were also prepared and measured under the same conditions. Based on the direct excitation from the X-ray tube, only the characteristic radiation of Nd, Gd, and Ti was detected, Figures 1–3. As illustrated in Figures 1–3, the characteristic Kα and Kβ lines of Ti were free from spectral interference and were detected at photon energies of 4.509 and 4.932 keV, respectively. Additional sum peaks of Ti were recognized at photon energies of 9.019 and 9.452 keV. Using the EDXRF spectra of the current nanocomposites, standard-less quantitative elemental analysis based on the fundamental parameter approach was performed. The proposed method is based on the conversion of the measured intensities (primary, secondary, and ternary) of each element of interest to concentrations in wt.%. Standard-less software "UniQuant" delivered from the manufacturer was utilized for this purpose. The current method has many advantages, such as how fast it is, its simplicity, complete absence of the standard materials, and automatic matrix correction. Figures 1 and 2 depict the X-ray fluorescence spectra of the $Gd_x/TiO_2$ and $Nd_x/TiO_2$ series at different concentrations of Gd and Nd ranging from 0.5 to 6 wt.%. The characteristic L lines of Gd were detected, and all of them are free from spectral interferences. The detected Gd-L lines are $L\alpha_1$, $L\beta_1$, $L\beta_2$, $L\gamma_1$, $L\gamma_2$, and $L_\iota$ at photon energies of 6.058, 6.714, 7.103, 7.786, 8.084, and 5.36 keV, respectively, Figures 1 and 3. The characteristic radiations of $Nd-L\alpha_1$, $Nd-L\beta_1$, $Nd-L\beta_2$ were recognized at 5.231, 5.722, and 6.09 keV, respectively, Figures 2 and 3. In the case of the $Gd_{0.5}/Nd_{0.5}/TiO_2$ NPs, there was a spectral interference between $Nd-L\beta_2$ at 6.09 keV and $Gd-L\alpha_1$, which was located at 6.058 keV, Figure 3. Additional spectral interferences between $Nd-L\gamma_1$ and $Gd-L\beta_1$ were observed at 6.602 and 6.714 keV, respectively. Although there was a remarkable spectral interference between L lines of Nd and Gd, the $Nd-L\alpha_1$ and $Nd-L\beta_1$ at, respectively, 5.231 and 5.722 keV were free from interference and can be used. In addition, the $Gd-L\beta_2$ at 7.103 keV and $Gd-L\gamma_1$ at 7.786 keV were also free from spectral interference. As shown in Figures 1 and 2, the characteristic L lines of Gd and Nd increase as the concentration of Gd and Nd increases. The quantitative elemental analysis results of the $Gd_x/TiO_2$ and $Nd_x/TiO_2$ NPs are presented in Table 1. The obtained concentrations (wt.%) of both $Gd_x/TiO_2$ and $Nd_x/TiO_2$ NPs are almost in good agreement with the initial prepared ratios. Table 2 summarizes the quantitative elemental analysis results of $Gd_{0.5}/TiO_2$, $Nd_{0.5}/TiO_2$, and $Gd_{0.5}/Nd_{0.5}/TiO_2$ NPs. In the case of $Gd_{0.5}/Nd_{0.5}/TiO_2$ NPs, the L lines of Gd and Nd are close to Ti-K lines and below 10 keV. The obtained results are in accordance with expected ratios.

X-ray diffraction (XRD) was employed to check the $Gd^{3+}$- and $Nd^{3+}$-doped $TiO_2$ phases and their crystallinity. Both $Gd_x/TiO_2$ and $Nd_x/TiO_2$ (x = 0.5, 1.0, 3.0, and 6.0 wt.%) samples only comprised diffraction peaks of the tetragonal anatase phase (JCPDS # 01-084-1286) [37]. No peaks for any oxide byproducts were observed for all the samples, most probably due to the XRD detection limit. This reflects the substitution of both Gd and Nd (up to 6%) in the $TiO_2$ anatase lattice forming a homogeneous solid solution. The structural information for all the refined phases was obtained by Rietveld refinements [38,39]. The anatase phase with the tetragonal lattice was refined in the space group $I4_1/amd$ [40] and dominated the composition of all $TiO_2$ samples (Table 1). Figure 4a–c shows the calculated

and observed diffraction patterns from Rietveld refinement for three selected samples where $R_{wp}$ (%) = 4 to 5.

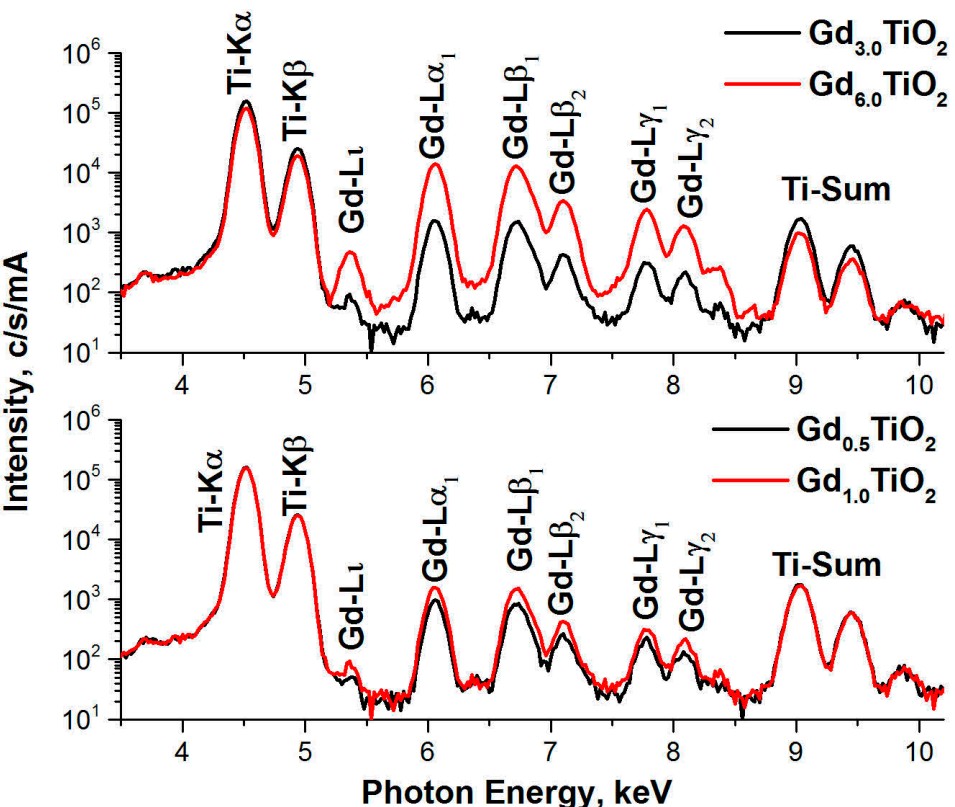

**Figure 1.** The EDXRF spectra of the $Gd_x/TiO_2$ nanocomposites.

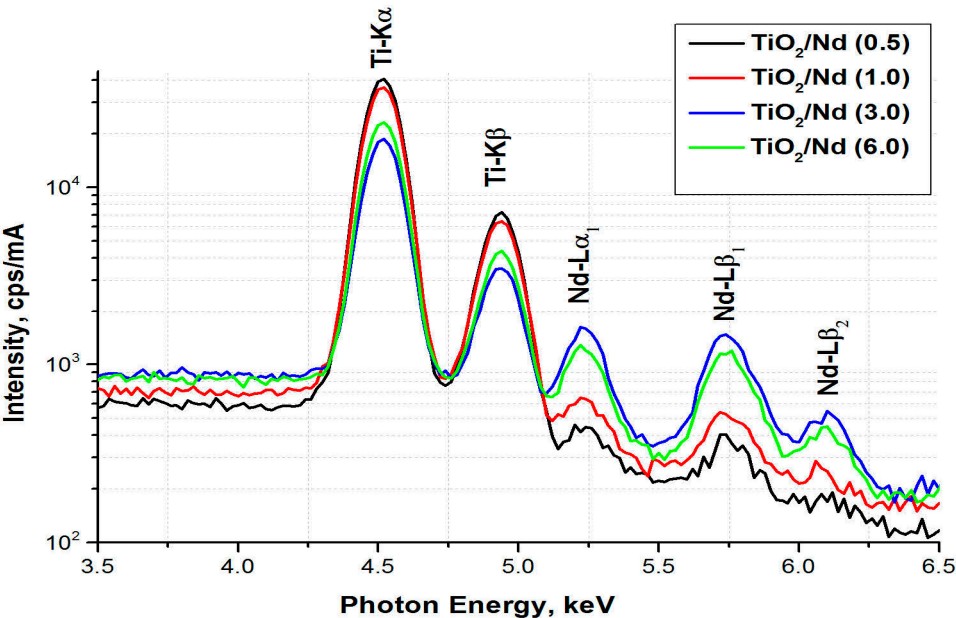

**Figure 2.** The EDXRF spectra of the $Nd_x/TiO_2$ nanocomposites.

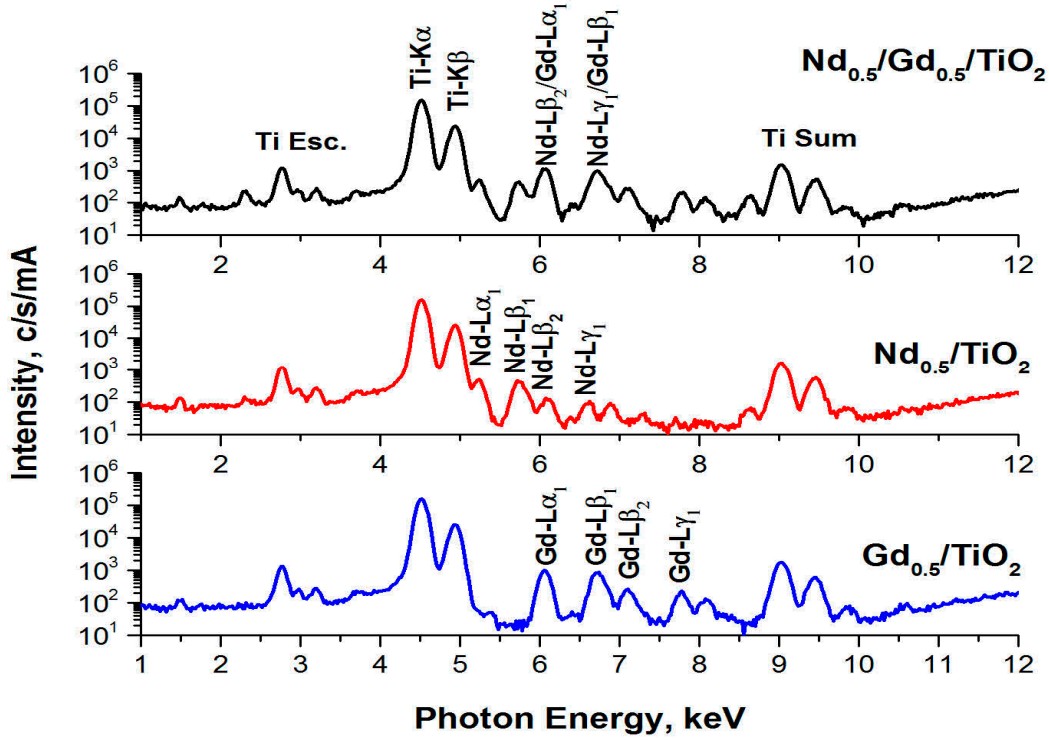

**Figure 3.** The EDXRF spectra of the $Gd_{0.5}/TiO_2$, $Nd_{0.5}/TiO_2$, and $Gd_{0.5}/Nd_{0.5}/TiO_2$ nanocomposites.

**Table 1.** Elemental quantitative analysis in wt.% of the $Gd_{xx}/TiO_2$ and $Nd_{xx}/TiO_2$ composites using Energy dispersive X-ray fluorescence (EDXRF) analysis.

| Gd$_x$/TiO$_2$ Composites | | | Nd$_x$/TiO$_2$ Composites | | |
|---|---|---|---|---|---|
| Sample | Gd, % | TiO$_2$, % | Sample | Nd, % | TiO$_2$, % |
| $Gd_{0.50}/TiO_2$ | $1.80 \pm 0.06$ | $97.92 \pm 0.07$ | $Nd_{0.5}/TiO_2$ | $1.13 \pm 0.05$ | $98.67 \pm 0.06$ |
| $Gd_{1.0}/TiO_2$ | $2.20 \pm 0.07$ | $97.46 \pm 0.08$ | $Nd_{1.0}/TiO_2$ | $2.16 \pm 0.07$ | $97.45 \pm 0.08$ |
| $Gd_{3.0}/TiO_2$ | $2.39 \pm 0.07$ | $97.24 \pm 0.08$ | $Nd_{3.0}/TiO_2$ | $7.71 \pm 0.12$ | $90.95 \pm 0.14$ |
| $Gd_{6.0}/TiO_2$ | $3.07 \pm 0.08$ | $96.45 \pm 0.09$ | $Nd_{6.0}/TiO_2$ | $10.89 \pm 0.25$ | $87.19 \pm 0.17$ |

**Table 2.** Elemental quantitative analysis in wt.% of the $Gd_{0.5}/TiO_2$, $Nd_{0.5}/TiO_2$ and $Gd_{0.5}/Nd_{0.5}/TiO_2$ composites using EDXRF.

| Sample | Gd$_{0.5}$/TiO$_2$ | Nd$_{0.5}$/TiO$_2$ | Gd$_{0.5}$/Nd$_{0.5}$/TiO$_2$ |
|---|---|---|---|
| Gd | $2.01 \pm 0.07$ | - | $2.11 \pm 0.07$ |
| Nd | - | $1.13 \pm 0.05$ | $1.04 \pm 0.05$ |
| TiO$_2$ | $97.85 \pm 0.07$ | $98.67 \pm 0.06$ | $96.35 \pm 0.09$ |

Figure 4d–f displays the tetragonal lattice parameters and the cell volume of the anatase phase as a function of Gd content. According to Vegard's law [41], the cell parameter *a* increased, which was attributed to the replacement of Ti with larger Gd atoms inside the anatase lattice. On the other hand, the cell parameter *c* decreased upon increasing the doping content and could be understood as a relaxation of the lattice due to the expansion of the *a* cell parameter. In turn, the volume of the anatase tetragonal cell was dominated by the enlargement of the *a* parameter and featured a slight increase with increasing the Gd and Nd content. In addition, the crystallite size was about 20 nm in the case of Gd doping, while doping with Nd reduced the crystallite size to about 10 nm, Table 3.

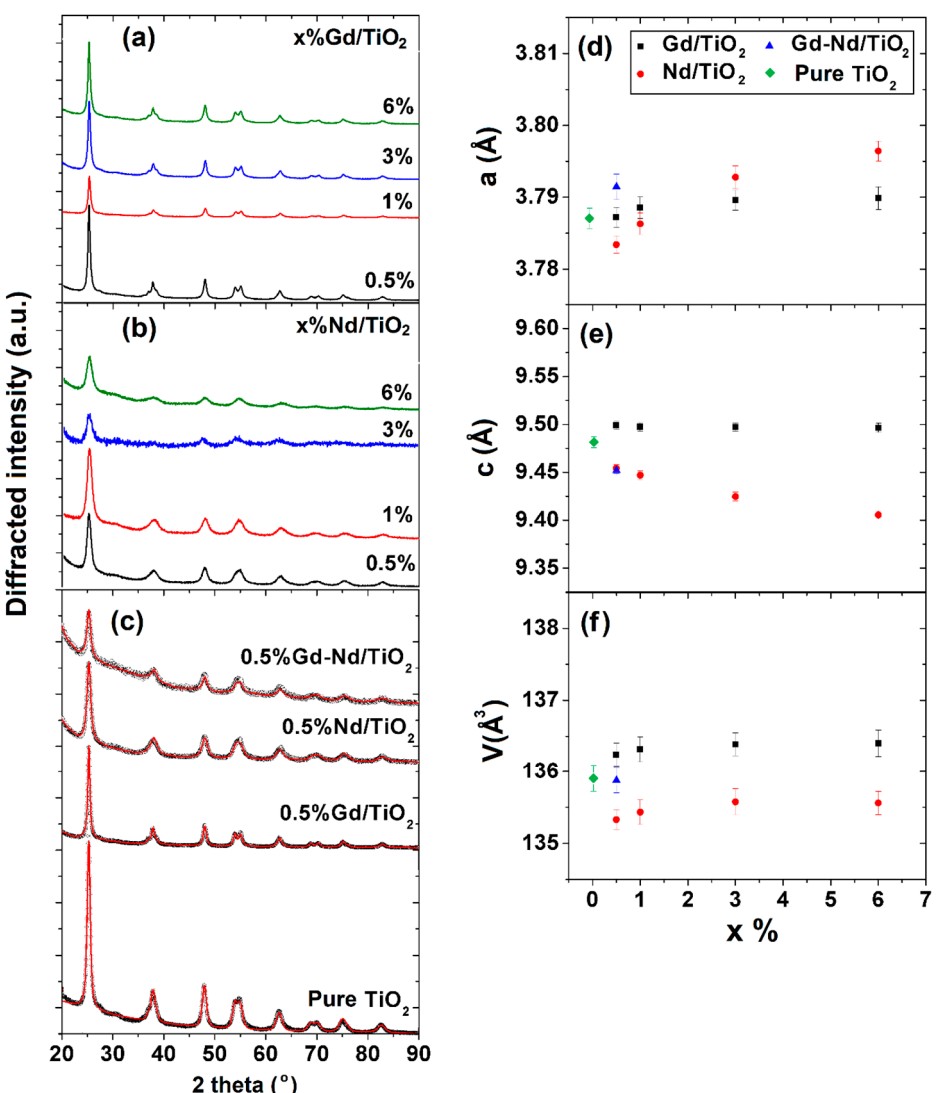

**Figure 4.** XRD patterns of (**a**) $Gd_x/TiO_2$ NPs and (**b**) $Nd_x/TiO_2$ NPs, (**c**) calculated (red) and recorded (black) diffraction patterns of $Gd_{0.5}/TiO_2$, $Nd_{0.5}/TiO_2$ and $Gd_{0.5}/Nd_{0.5}/TiO_2$, (**d**,**e**) lattice parameters as a function of Gd and Nd content, (**f**) cell volume as a function of Gd and Nd content.

**Table 3.** The cell parameters and crystallite size of $Gd_x/TiO_2$, $Nd_x/TiO_2$, and $Gd_{0.5}/Nd_{0.5}/TiO_2$ NPs as calculated from the Rietveld refinements.

| x | $Gd_x/TiO_2$ | | | | $Nd_x/TiO_2$ | | | |
|---|---|---|---|---|---|---|---|---|
| | a (A) | c (A) | V (A³) | D (nm) | a (A) | c (A) | V (A³) | D (nm) |
| 0.5 | 3.7872 (14) | 9.499 (4) | 136.24 (16) | 21 (2) | 3.7834 (12) | 9.454 (4) | 135.33 (14) | 12 (1) |
| 1.0 | 3.7885 (15) | 9.497 (5) | 136.31 (17) | 19 (2) | 3.7863 (15) | 9.447 (5) | 135.43 (17) | 11 (1) |
| 3.0 | 3.7895 (14) | 9.497 (4) | 136.39 (16) | 18 (1) | 3.7928 (16) | 9.425 (5) | 135.58 (18) | 10 (2) |
| 6.0 | 3.7899 (16) | 9.496 (5) | 136.40 (18) | 18 (1) | 3.7964 (14) | 9.405 (4) | 135.56 (16) | 10 (2) |
| | 0.5%Gd-0.5%Nd/$TiO_2$ | | | | Pure $TiO_2$ | | | |
| | 3.7915 (18) | 9.452 (3) | 135.87 (17) | 12 (2) | 3.7854 (14) | 9.4842 (5) | 135.9 (17) | 17 (3) |

## 2.2. Composition and Chemical State Analysis

The composition and chemical state of pure $TiO_2$, $Gd_x/TiO_2$, $Nd_x/TiO_2$, and $Gd_{0.5}/Nd_{0.5}/TiO_2$ NPs were characterized by electron spectroscopy for chemical analysis (ESCA) set up using a monoenergetic Al-Kα (1486.6 eV). Figure 5 exhibits the survey spectrum of the

pure TiO$_2$ nanoparticles as well as the core level spectra of the Ti 2p and O 1s. The core levels plot of Ti and O were recognized not only in the pure TiO$_2$, but also in the doped TiO$_2$, Figures 5–7. The peak of the C 1s at 284.6 eV as well as the KVV Auger line at 1233 eV originate from the adsorption of carbon on the surface due to contamination. The C 1s peak was used as a reference to correct the charge shift in the doped TiO$_2$ [42]. Two principle peaks of Ti 2p at the binding energies of 458.8 and 464.4 eV were allocated to Ti 2p$_{3/2}$ and Ti 2p$_{1/2}$, respectively. Besides, the Ti Auger lines were assigned to LM$_{23}$M$_{23}$ and L$_3$M$_{23}$M$_{45}$ at 1103 and 1073 eV, respectively. The spin-orbit splitting of 5.7 eV between the two peaks of Ti 2p confirms the existence of titanium dioxide [43]. Therefore, the oxidation state of titanium is mostly +4, and this is consistent with the literature data.

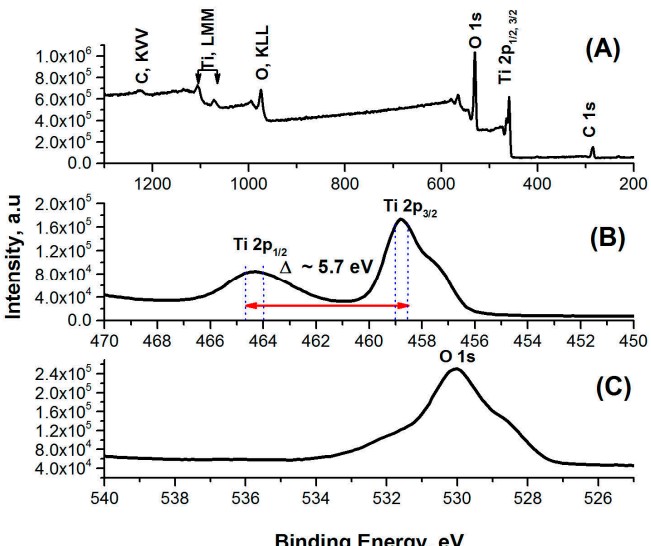

**Figure 5.** X-ray photoelectron spectroscopy analysis of the pure TiO$_2$ nanoparticles including (**A**) a survey and core level spectra of (**B**) Ti 2p and (**C**) O 1s.

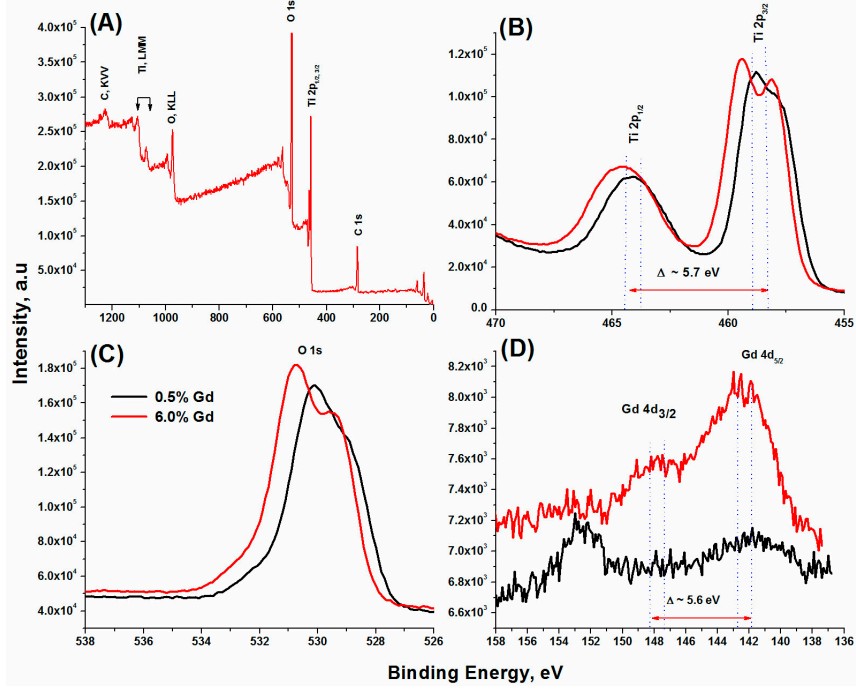

**Figure 6.** X-ray photoelectron spectroscopy of the Gd/TiO$_2$ nanoparticles at 0.5% and 6.0% of Gd including (**A**) a survey and core level spectra of (**B**) Ti 2p, (**C**) O 1s, and (**D**) Gd 4d.

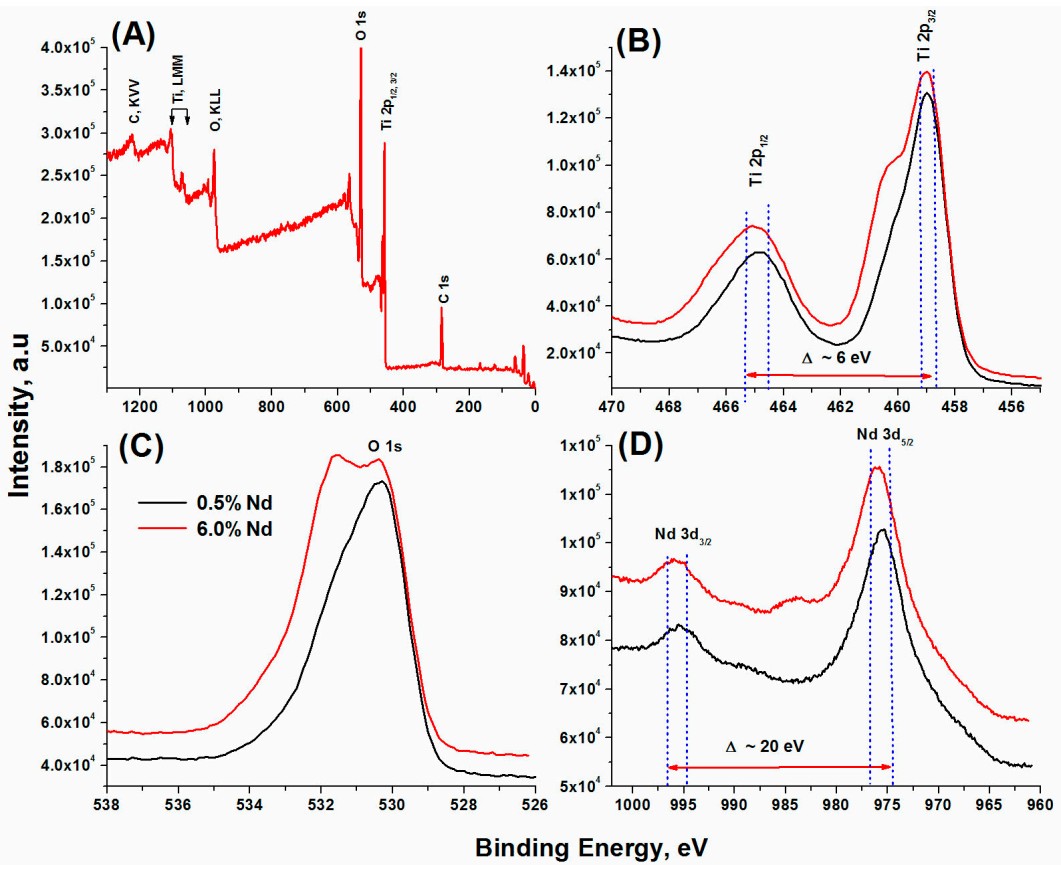

**Figure 7.** X-ray photoelectron spectroscopy of the Nd-doped TiO$_2$ nanoparticles at 0.5 and 6.0 wt.% of Nd including (**A**) a survey and core spectra of (**B**) Ti 2p, (**C**) O 1s, and (**D**) Nd 3d.

The shifts of the Ti 2p to high binding energy with Gd and Nd could be attributed to either new species of the TiO$_2$ or the inelastic scattering of Gd and Nd atoms within the anatase crystal structure [44].

The peak of O 1s was estimated at 530.1 eV at the low content of doping (0.5%). At the high doping level (6.0%) of Gd and Nd, two peaks of O 1s were observed at 529.8 and 530.7 eV, respectively, Figures 6 and 7. The different oxygen species could be the main reason for the shoulder at higher binding energy. Additional KLL Auger lines could be recognized at around 1000 eV. The two peaks of O 1s in the doped TiO$_2$ were attributed to the crystal lattice oxygen at 529.8 eV and hydroxyl oxygen (Ti-OH) for pure TiO$_2$ at 530.7 eV, Figures 6 and 7 [45]. However, the different oxygen species could be the main reason for the shoulder at higher binding energy. As the concentrations of Gd and Nd increase from 0.5% to 6% in the anatase phase, remarkable shifts of the O 1s peak was observed, which suggests the presence of Gd$_2$O$_3$ and Nd$_2$O$_3$ in the anatase crystal [46,47]. Additionally, the remarkable shifts of the O 1s peak could be attributed to the surface relaxation effects [48].

In the case of Gd-doped TiO$_2$, two weak peaks of Gd were identified at 142 and 152 eV ascribed to 4p3/2, 4d5/2, respectively, Figure 6. Due to the low detector efficiency at high Z elements, the statistical distribution of the Gd 4d peaks was poor, and the Gd 3d peaks could not be detected at a low concentration of Gd (0.5%). The binding energy of Gd 4d$_{5/2}$ at 142.4 eV indicates the trivalent oxidation state of gadolinium, mostly in the form of Gd$_2$O$_3$, which is in agreement with reference [49].

The spin orbit splitting between the two peaks of Gd 4d equals 5.6 eV, and it also agrees with the literature [49]. Based on the obtained spectra of Gd$_x$/TiO$_2$ NPs, the characteristic peaks of Gd were successfully evidenced in the TiO$_2$, Figure 6. The valence state of Nd$_x$/TiO$_2$ NPs was also demonstrated. The complete doping behavior was also confirmed

in the case of Nd $_x$/TiO$_2$ NPs. The two characteristic peaks of Nd 3d$_{3/2}$ and Nd 3d$_{5/2}$ were located, respectively, at 995.8 and 975.9 eV, Figure 7. The values of the binding energy of Nd 3d agree with those reported by Wang et al. [50]. The spin orbit splitting between the two peaks of Nd 3d equals 20 eV, Figure 7.

As seen from Figures 5–8, the intensity of O 1s peak decreased after doping with Gd or Nd compared with the pure TiO$_2$, which confirms the successful doping in TiO$_2$. In the case of Gd$_x$/TiO$_2$ and Nd$_x$/TiO$_2$ NPs, the binding energies (BE) of the two peaks of Ti 2p$_{1/2}$ and 2p$_{3/2}$ at 464.8 and 458.9 eV, respectively, are lower than those of pure TiO$_2$. Besides, the BE of the two peaks of Ti in Gd$_x$/TiO$_2$ are also lower than those of Nd$_x$/TiO$_2$ NPs, Figures 5–8. The decrease in Ti 2p BE might be attributed to the Ti$^{4+}$ and O$^{2-}$ local environment change by the introduction of Gd or Nd atoms.

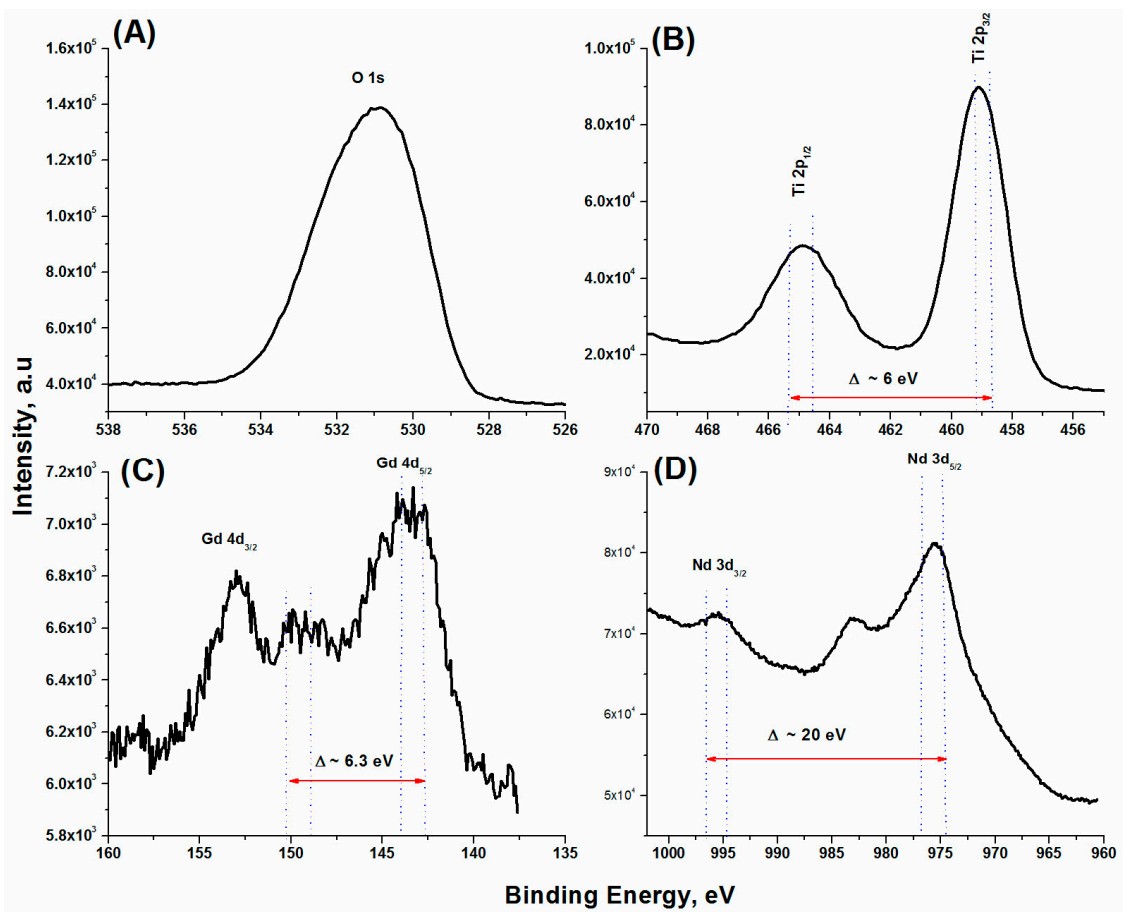

**Figure 8.** XPS core level spectra of (**A**) O 1s, (**B**) Ti 2p, (**C**) Gd 4d, and (**D**) Nd 3d of the Gd$_{0.5}$/Nd$_{0.5}$/TiO$_2$ NPs.

As shown in Figures 6 and 7, these shifts in the BE of Ti 2p and O 1s peaks could be ascribed to the formation of Ti –Gd and Ti–Nd bonds on the grain boundaries of the crystallites, reducing the Ti$^{4+}$ BE. Figures 6 and 7 revealed an increase of Nd 3d and Gd 4d peak intensities upon increasing the doping level from 0.5 to 6.0 wt.%. At a concentration of 0.5 wt.% of Nd and Gd, the O 1s and Ti 2p intensities had a remarkable decrease. The spin orbit splitting of Nd 3d and Gd 4d remained the same as the previous cases.

Tables 4–7 summarize the measured binding energies, full widths at half maximum (FWHM), peak areas, and atomic concentrations for pure TiO$_2$, Gd$_x$/TiO$_2$ and Nd$_x$/TiO$_2$ NPs, and Gd$_{0.5}$/Nd$_{0.5}$/TiO$_2$ NPs. The non-stoichiometric atomic ratios of O:Ti of TiO$_2$ could be ascribed to two main reasons. The first one is the quantitative surface analysis capability of the XPS is limited to a few nanometers (1–10 nm). Secondly, due to the chemical treatments of TiO$_2$, Gd$_x$/TiO$_2$ and Nd$_x$/TiO$_2$ NPs, and Gd$_{0.5}$/Nd$_{0.5}$/TiO$_2$ NPs [51], the

atomic ratios of O:Ti of TiO$_2$ could be relatively non-stoichiometric, especially at the surface of the NPs. As illustrated in Tables 4–6, the O:Ti atomic ratios in Nd$_x$/TiO$_2$ samples are higher than those of Gd$_x$/TiO$_2$. These results support the reported fact that Nd$^{3+}$ ions are more electropositive than Gd$^{3+}$ ions. Therefore, the Nd doping may create a more oxygen-rich nano-phase structure.

**Table 4.** The measured binding energies, full widths at half maximum (FWHM), peak areas, and atomic concentrations for the pure TiO$_2$ NPs.

| Peak | Binding Energy (eV) | FWHM (eV) | Peak Area, kcps (eV) | Atomic Conc. (at.%) |
|---|---|---|---|---|
| O 1s | 530.77 | 7.49 | 3398.3 | 73.4 |
| Ti2p$_{1/2}$ Ti2p$_{3/2}$ | 458.29 463.98 | 4.74 | 2892.5 | 26.5 |

**Table 5.** The measured binding energies, full widths at half maxima (FWHM), peak areas, and atomic concentrations for the Gd/TiO$_2$ NPs.

| Peak | Binding Energy (eV) | FWHM (eV) | Peak Area, kcps (eV) | Atomic Conc. (at.%) |
|---|---|---|---|---|
| | | Gd$_{0.5}$/TiO$_2$ | | |
| O 1s | 530.77 | 3.6 | 991.5 | 73.43 |
| Ti 2p$_{1/2}$ Ti 2p$_{3/2}$ | 458.29 463.98 | 3.31 | 973.6 | 26.48 |
| Gd 4d$_{3/2}$ Gd 4d$_{5/2}$ | 142 152 | 0.27 | 1.3 | 0.09 |
| | | Gd$_{6.0}$/TiO$_2$ | | |
| O 1s | 530.77 | 3.71 | 1047.5 | 71.73 |
| Ti 2p$_{1/2}$ Ti 2p$_{3/2}$ | 458.29 463.98 | 3.23 | 1049.4 | 28.11 |
| Gd 4d$_{3/2}$ Gd 4d$_{5/2}$ | 142 152 | 0.0 | 1.3 | 0.16 |

**Table 6.** The measured binding energies, full widths at half maxima (FWHM), peak areas, and atomic concentrations for the Nd-doped TiO$_2$ nanoparticles.

| Peak | Binding Energy (eV) | FWHM (eV) | Peak Area, kcps (eV) | Atomic Conc. (at.%) |
|---|---|---|---|---|
| | | Nd (0.5%)/TiO$_2$ | | |
| O 1s | 530.77 | 3.36 | 1139.1 | 75.12 |
| Ti 2p$_{1/2}$ Ti 2p$_{3/2}$ | 458.29 463.98 | 1.93 | 1108.7 | 24.71 |
| Nd 3d$_{3/2}$ Nd 3d$_{5/2}$ | 995.8 975.9 | 0 | 0.774 | 0.17 |
| | | Nd (6.0%)/TiO$_2$ | | |
| O 1s | 530.77 | 3.7 | 1266.5 | 74.59 |
| Ti 2p$_{1/2}$ Ti 2p$_{3/2}$ | 458.29 463.98 | 3.37 | 1175.9 | 25.18 |
| Nd 3d$_{3/2}$ Nd 3d$_{5/2}$ | 975.9 995.8 | 0.01 | 1.78 | 0.23 |

**Table 7.** The measured binding energies, full widths at half maxima (FWHM), peak areas, and atomic concentrations for the $Gd_x/TiO_2$ and $Nd_x/TiO_2$ NPs.

| Peak | Binding Energy, eV | FWHM, eV | Peak Area, kcpseV | Atomic Concentration, (at.%) |
|---|---|---|---|---|
| O 1s | 538.5 | 3.83 | 961.4 | 77.55 |
| Ti $2p_{1/2}$<br>Ti $2p_{3/2}$ | 458.29<br>463.98 | 2.89 | 657.9 | 21.82 |
| Gd $4d_{3/2}$<br>Gd $4d_{5/2}$ | 142<br>152 | 0.19 | 0.70 | 0.29 |
| Nd $3d_{3/2}$<br>Nd $3d_{5/2}$ | 975.9<br>995.8 | 0.64 | 3.8 | 0.34 |

SEM imaging and EDX were used jointly to elucidate the morphology and elemental composition of the $Gd_x/TiO_2$ and $Nd_x/TiO_2$ NPs. Figure 9 shows the SEM photographs of the typical $Gd_x/TiO_2$ and $Nd_x/TiO_2$ samples. From the images, the $Gd_x/TiO_2$ and $Nd_x/TiO_2$ existed essentially in the form of spherical particles and presented porous structures similar to those of $TiO_2$. The morphological study revealed that for both $TiO_2$ and $Gd_x/TiO_2$ samples, the surface looked almost the same with slightly whitish portion, indicating the deposition of Gd. Based on the SEM results, the Ti K$\alpha$-fluorescence signals of the pure $TiO_2$ and $Gd_x/TiO_2$ samples were also obtained by EDX analysis (Figure 9). Table 8 gives semi-qualitative information about the elemental and atomic percentages in the $TiO_2$ and $Gd_x/TiO_2$ samples.

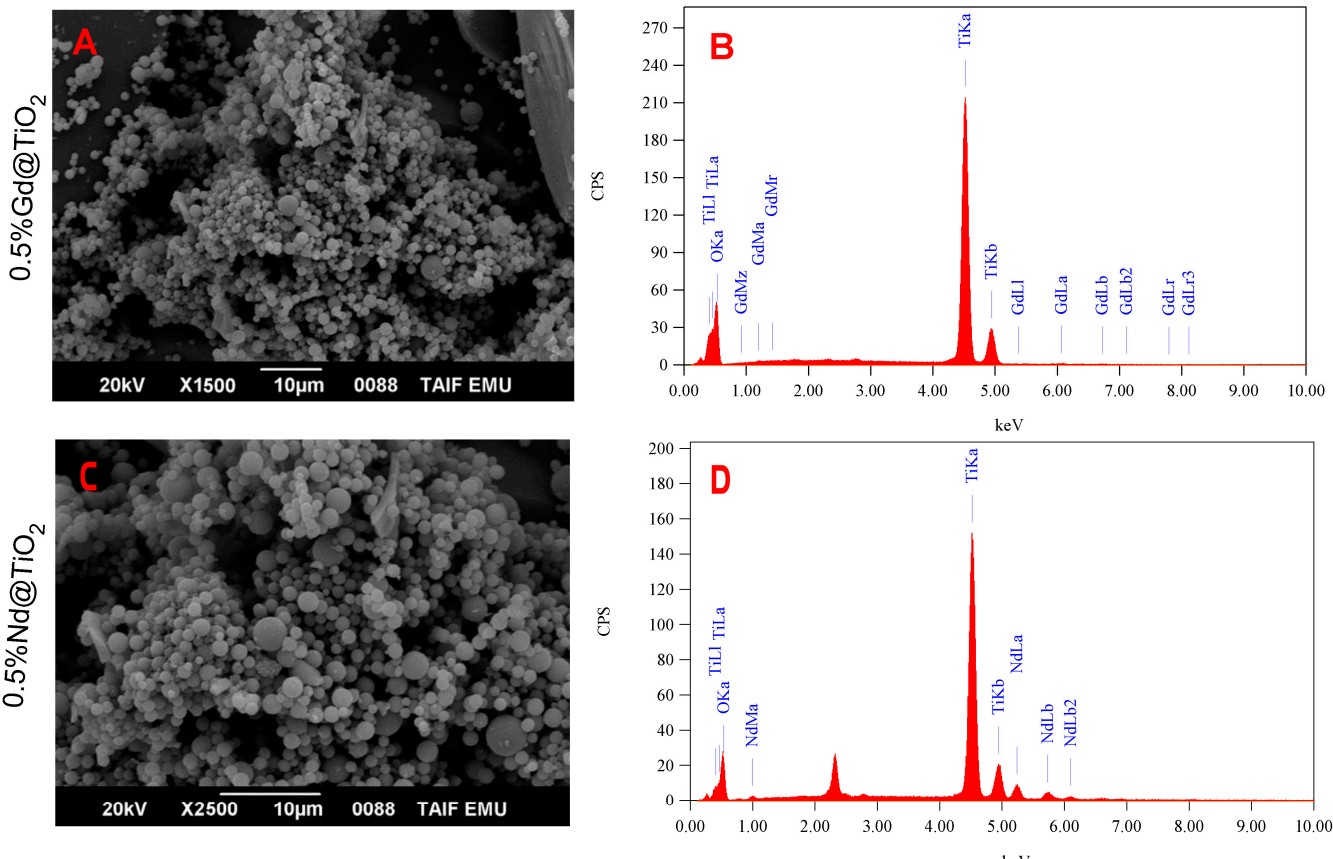

**Figure 9.** SEM/EDX examinations of $Gd_{0.5}/TiO_2$ (**A,B**) and $Nd_{0.5}/TiO_2$ NPs (**C,D**).

**Table 8.** Catalyst composition using EDX analysis of $TiO_2$, $Gd/TiO_2$, and $Nd/TiO_2$.

| Element | Compound | | |
|---|---|---|---|
| | $TiO_2$ (Atom%) | $Gd_x/TiO_2$ (Atom%) | $Nd_x/TiO_2$ (Atom%) |
| O | 73.04 | 70.81 | 72.36 |
| Ti | 26.96 | 29.05 | 27.46 |
| Gd | - | 0.150 | - |
| Nd | - | - | 0.185 |

High Resolution Transmission Electron Microscopy (HR-TEM) Analysis

The morphology of the samples and the corresponding chemical composition were further determined by, respectively, HR-TEM and selected area electron diffraction (SAED) patterns. Figure 10 depicts typical TEM and HR-TEM images (insets SAED patterns) of $Nd_{0.5}/TiO_2$, $Gd_{0.5}/TiO_2$, and $Gd_{0.5}/Nd_{0.5}/TiO_2$ NPs. As evidenced in the TEM images, the majority of the $TiO_2$ nanoparticles consist mainly of quasi-spherical and cubic particles. The HR-TEM images revealed a characteristic lattice spacing of 0.352 nm for the $TiO_2$ anatase (101) plane. The average size of both $Gd_{0.5}/TiO_2$ and $Nd_{0.5}/TiO_2$ slightly changed with the increase of the $RE^{3+}$ content; for example, the average particle size range was 10–12 nm.

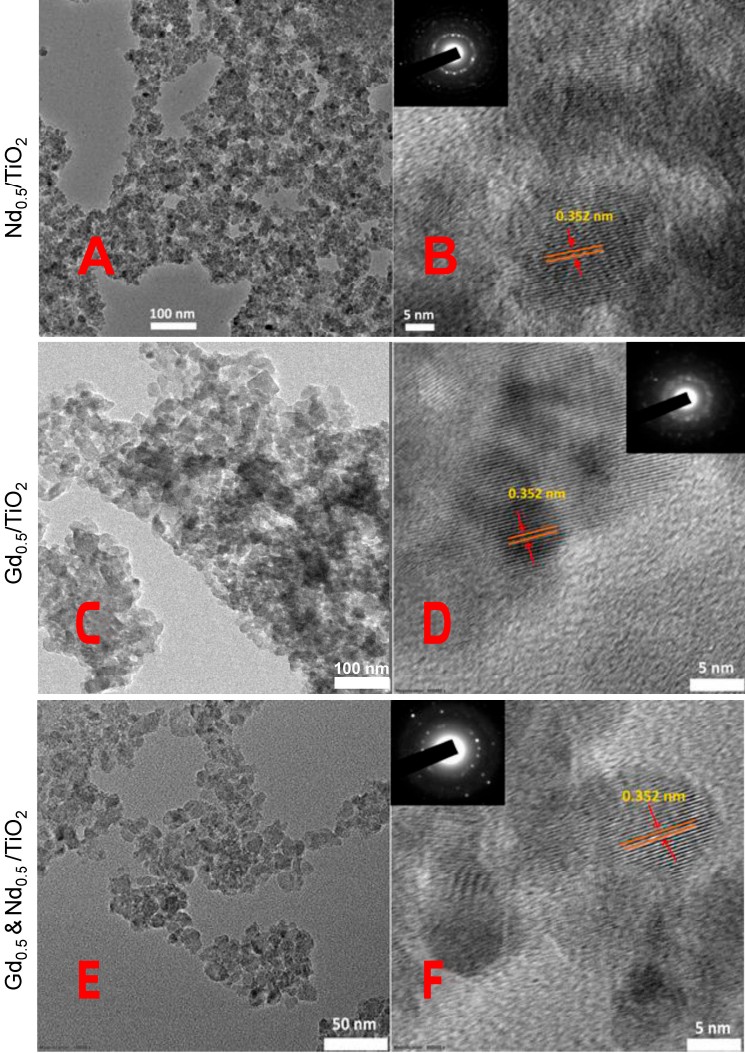

**Figure 10.** Transmission electron microscopy (TEM) and high-resolution TEM (HR-TEM) images (inset SAED patterns) of $Gd_{0.5}/TiO_2$, $Nd_{0.5}/TiO_2$, and $Gd_{0.5}/Nd_{0.5}/TiO_2$ (average particle size 12 ± 0.50 nm). (**A,B**) $Gd_{0.5}/TiO_2$, (**C,D**) $Nd_{0.5}/TiO_2$, and (**E,F**) $Gd_{0.5}/Nd_{0.5}/TiO_2$.

To estimate the optical band gap, UV-vis diffuse reflectance spectra were measured to analyze the red-shifts in the absorption regions. The Kubelka–Munk equation $\alpha h\nu = A(h\nu - E_g)^2$, where $\alpha$, $h$, $\nu$, and $E_g$ and $A$ are the absorption coefficient, Plank constant, light frequency, band gap, and the proportionality constant, respectively, was used for band gap determination, Figures 11 and 12. For pure TiO$_2$, a band gap energy of 3.18 eV was determined, which is in accordance with that of other reports [52–54], while for RE-doped TiO$_2$, the band gap energy decreased due to the red-shift of absorbance (see Figures 11 and 12 and Table 9), suggesting that gadolinium and neodymium improved the visible light absorbance of TiO$_2$.

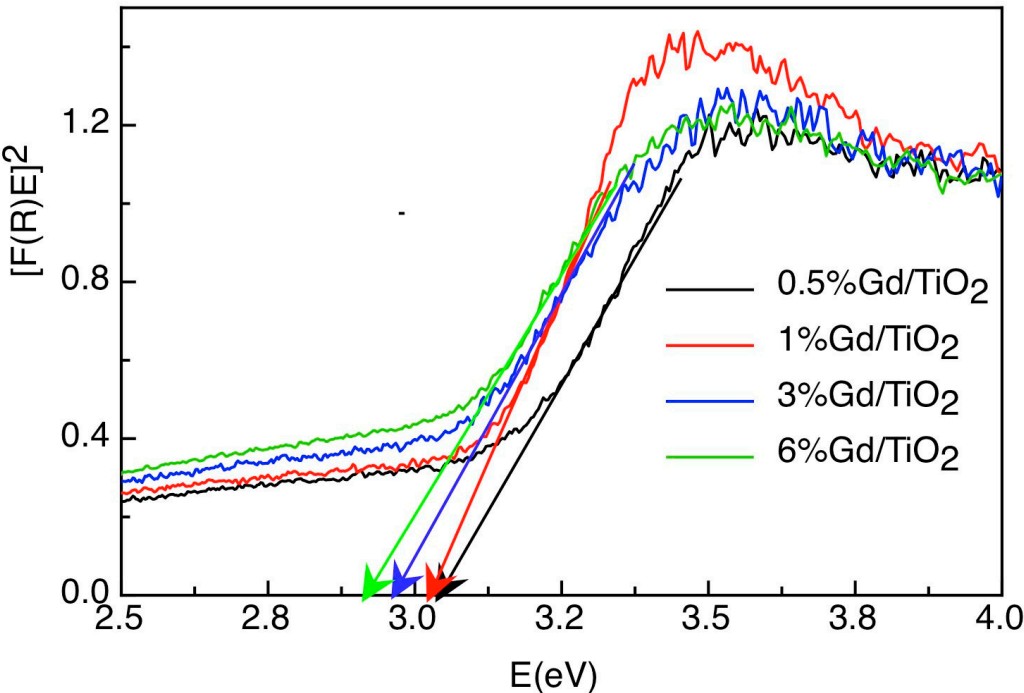

**Figure 11.** Kubelka–Munk function plots of Gd$_x$/TiO$_2$ NPs for different Gd atomic concentrations.

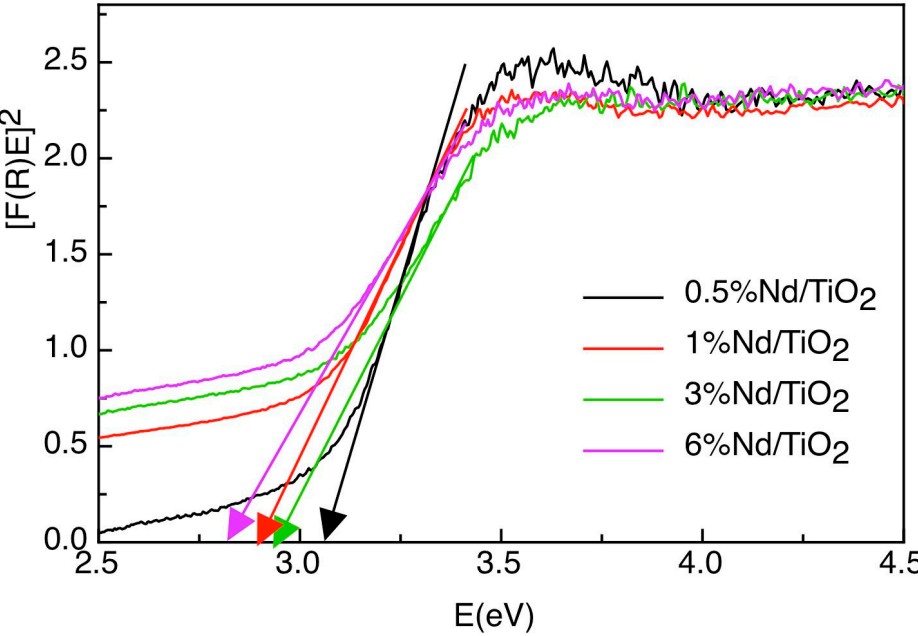

**Figure 12.** Kubelka–Munk function plots of Nd$_x$/TiO$_2$ NPs for different Nd atomic concentrations.

**Table 9.** Band gaps of $Gd_x/TiO_2$ and $Nd_x/TiO_2$ NPs with different atomic concentrations.

|  | 0.5% | 1.0% | 3.0% | 6.0% |
|---|---|---|---|---|
| $Gd_x/TiO_2$ | 3.06 | 3.04 | 2.97 | 2.93 |
| $Nd_x/TiO_2$ | 3.07 | 2.95 | 2.91 | 2.83 |

*2.3. Electrocatalytic Activity Studies for the Hydrogen Evolution Reaction (HER)*

2.3.1. Cathodic Polarization Measurements

Figure 13a presents the cathodic polarization plots of our synthesized catalysts, namely, $Gd_x/TiO_2$ and $Nd_x/TiO_2$ with various RE doping percentages (0.5, 1, 3, and 6%). The cathodic polarization curves of the $Gd_{1.0}/TiO_2$ NPs with varying Nd content, $Gd_{1.0}/Nd_x/TiO_2$ (x = 0.5, 1, 3, and 6%) were also recorded. Measurements were conducted in 1.0 M KOH solution in comparison with bare GCE and $TiO_2/GCE$. The polarization curves in Figure 13a also comprised the cathodic response of a commercial Pt/C catalyst as a reference point. The corresponding Tafel plots are exhibited in Figure 13b, and the fitting Tafel parameters are depicted in Table 10.

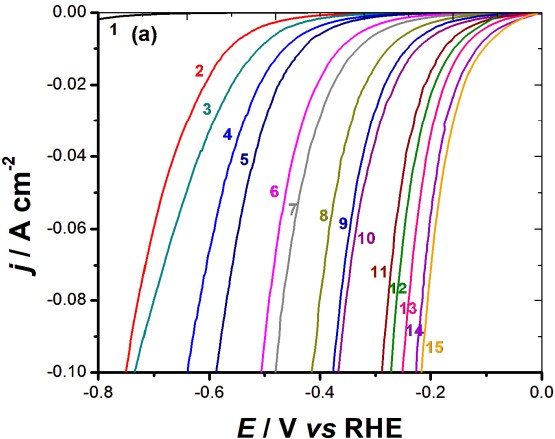

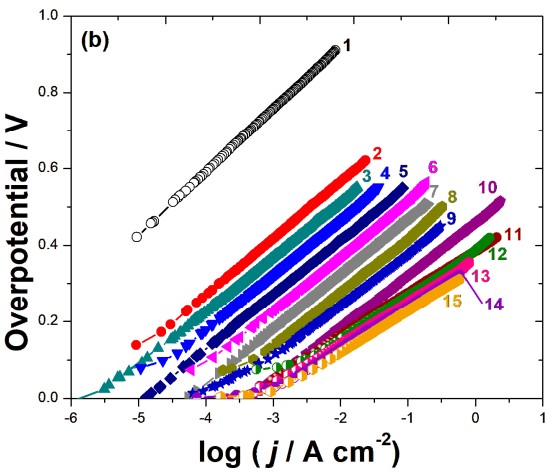

**Figure 13.** Cathodic polarization measurements for (**a**) the HER and (**b**) the corresponding Tafel plots recorded for the investigated catalysts. Measurements were carried out in 1.0 M KOH solution at a scan rate of 5 mV s$^{-1}$ at room temperature. (1) bare GCE; (2) $TiO_2/GCE$; (3) $Gd_{0.5}/TiO_2/GCE$; (4) $Gd_{1.0}/TiO_2/GCE$; (5) $Gd_{3.0}/TiO_2/GCE$; (6) $Gd_{6.0}/TiO_2/GCE$; (7) $Nd_{0.5}/TiO_2/GCE$; (8) $Nd_{1.0}/TiO_2/GCE$; (9) $Nd_{3.0}/TiO_2/GCE$; (10) $Nd_{6.0}/TiO_2/GCE$; (11) $Gd_{1.0}/Nd_{0.5}/TiO_2/GCE$; (12) $Gd_{1.0}/Nd_{1.0}/TiO_2/GCE$; (13) $Gd_{1.0}/Nd_{3.0}/TiO_2/GCE$; (14) $Gd_{1.0}/Nd_{6.0}/TiO_2/GCE$; (15) Pt/C.

**Table 10.** Mean value (standard deviation) of the electrochemical HER kinetic parameters on the surfaces of our synthesized catalysts, $Gd_x/TiO_2$, $Nd_x/TiO_2$, and $Gd_{1.0}/Nd_x/TiO_2$ loaded on a GCE. Measurements were conducted at room temperature in deaerated KOH solution (1.0 M) in comparison with bare GCE, $TiO_2/GCE$, and Pt/C.

| Tested Cathode | Onset Potential ($E_{HER}$, mV vs. RHE) | Tafel Slope ($\beta_c$, mV dec$^{-1}$) | Exchange Current Density ($j_o$, mA cm$^{-2}$) | Overpotential at $j$ = 10 mA cm$^{-2}$ ($\eta_{10}$, mV) |
|---|---|---|---|---|
| bare GCE | −720 (9.2) | −165 (2.6) | 2.75 (0.05) × 10$^{-5}$ | ---- |
| $TiO_2$/GCE | −215 (3.6) | −152 (2.2) | 1.45 (0.03) × 10$^{-3}$ | 565 (7.6) |
| $Gd_{0.5}/TiO_2$/GCE | −186 (3.2) | −142 (1.7) | 2.51 (0.04) × 10$^{-3}$ | 511 (6.2) |
| $Gd_{1.0}/TiO_2$/GCE | −175 (2.9) | −143 (1.8) | 5.4 (0.15) × 10$^{-3}$ | 466 (5.1) |
| $Gd_{3.0}/TiO_2$/GCE | −160 (2.8) | −141 (2.1) | 1.12 (0.3) × 10$^{-2}$ | 432 (4.7) |
| $Gd_{6.0}/TiO_2$/GCE | −145 (2.9) | −142 (1.8) | 2.51 (0.04) × 10$^{-2}$ | 365 (4.2) |
| $Nd_{0.5}/TiO_2$/GCE | −130 (2.2) | −140 (1.5) | 5.62 (0.06) × 10$^{-2}$ | 332 (3.8) |
| $Nd_{1.0}/TiO_2$/GCE | −118 (2.4) | −139 (1.5) | 10.5 (0.3) × 10$^{-2}$ | 277 (3.5) |
| $Nd_{3.0}/TiO_2$/GCE | −103 (1.5) | −141 (1.6) | 15.9 (0.42) × 10$^{-2}$ | 244 (3.2) |
| $Nd_{6.0}/TiO_2$/GCE | −85 (1.4) | 140 (1.5) | 43.6 (0.6) × 10$^{-2}$ | 225 (2.2) |
| $Gd_{1.0}/Nd_{0.5}/TiO_2$/GCE | −72 (1.6) | 113 (1.4) | 35.5 (0.4) × 10$^{-2}$ | 177 (1.9) |
| $Gd_{1.0}/Nd_{1.0}/TiO_2$/GCE | −60 (1.1) | 112 (1.4) | 44.7 (0.6) × 10$^{-2}$ | 161 (1.8) |
| $Gd_{1.0}/Nd_{3.0}/TiO_2$/GCE | −38 (0.8) | 110 (1.8) | 50.2 (0.7) × 10$^{-2}$ | 142 (1.7) |
| $Gd_{1.0}/Nd_{6.0}/TiO_2$/GCE | −22 (0.3) | 109 (1.5) | 72 (1.1) × 10$^{-2}$ | 115 (1.8) |
| Pt/C | −15 (0.2) | −106 (1.2) | 80 (0.9) × 10$^{-2}$ | 106 (1.5) |

As a well-known eminent HER electrocatalyst, the Pt/C catalyst achieved amongst the investigated catalysts the lowest HER's onset potential, $E_{HER}$~−15 mV vs. RHE, with the steepest reduction (catalytic) currents. In contrast, the bare GC electrode displayed inferior catalytic activity, clear from its humble catalytic current generated at a larger $E_{HER}$ (−720 mV vs. RHE).

On the other hand, there was a substantial improvement in the HER catalytic activity in the reductive sweep curves of the synthesized $Gd_x/TiO_2$, $Nd_x/TiO_2$, and $Gd_{1.0}/Nd_x/TiO_2$ nanocomposites. This enhanced HER catalytic activity occurred to different extents depending on the type of studied catalyst, RE doping percentage and the bimetallic nanocomposite $Gd_{1.0}/Nd_x/TiO_2$ composition.

It follows, from Figure 13a, that the HER catalytic activity of $Nd_x/TiO_2$ and $Gd_x/TiO_2$ catalysts enhanced with RE doping percentage. In addition, at any RE doping percentage, the $Nd_x/TiO_2$ electrocatalyst exhibited higher HER activity than $Gd_x/TiO_2$. This was evident from their $E_{HER}$ values recorded in Table 10. For instance, at a doping percentage of 3.0%, an $E_{HER}$ value of −103 mV vs. RHE was recorded for $Nd_{3.0}/TiO_2$, which is 57 mV more anodic (active direction) than that of the $Gd_{3.0}/TiO_2$ catalyst (160 mV vs. RHE). The lower $E_{HER}$ values of the $Nd_x/TiO_2$ catalysts led to higher exchange current density values, $j_o$. For example, a $j_o$ value of 15.9 × 10$^{-2}$ mA cm$^{-2}$ was recorded for the $Nd_{3.0}/TiO_2$ catalyst. This $Nd_{3.0}$-$TiO_2$ catalyst's $j_o$ value is ~14.2 times greater than that measured for the $Gd_{3.0}/TiO_2$ catalyst (1.12 × 10$^{-2}$ mA cm$^{-2}$). The higher HER catalytic activity of $Nd_x/TiO_2$ catalysts was also testified from their lower overpotentials required to generate a current density of 10 mA cm$^{-2}$, $\eta_{10}$. For example, the $Nd_{3.0}/TiO_2$ catalyst required an $\eta_{10}$ of 244 mV to deliver a current density of 10 mA cm$^{-2}$, which is 188 mV anodic to that attained by $Gd_{3.0}/TiO_2$ (432 mV).

These results highlight the high HER catalytic efficiency of the $Nd_x/TiO_2$ catalyst. The high $Nd_x/TiO_2$ electrocatalyst's HER activity compared to that of the $Gd_x/TiO_2$

electrocatalysts can be chiefly attributed to, as evidenced from XRD studies (revisit Table 3), the former's smaller crystallite size (~10 nm) than that of the latter (~20 nm). The $Nd_x/TiO_2$ electrocatalyst's smaller crystallite size is translated into higher electrochemical active surface area (EASA), as estimated from cyclic voltammetry measurements performed at various potential scan rates (Figure S3, Supporting Information).

The HER catalytic performance of the investigated $Gd_{1.0}/Nd_x/TiO_2$ bimetallic NPs (x = 0.5, 1, 3, and 6%) is positioned far beyond that of their corresponding individuals at any studied RE doping percentage, namely, $Gd_x/TiO_2$ and $Nd_x/TiO_2$ (x = 0.5, 1, 3, and 6%). In addition, the $Gd_{1.0}/Nd_x/TiO_2$ bimetallic NPs' catalytic activity for the HER was enhanced with increasing Nd content, approaching that of the commercial Pt/C electrocatalyst at a composition of $Gd_{1.0}/Nd_{6.0}/TiO_2$. This was evident from the $E_{HER}$ and $\eta_{10}$ values recorded for the tested $Gd_{1.0}/Nd_x/TiO_2$ bimetallic NPs, Table 10, that shift towards a more anodic direction with an increase in the Nd content. Thus, high cathodic currents, and hence large amounts of $H_2$, could be generated at low overpotentials, denoting efficacious catalytic performance for the HER.

Increasing the Nd doping percentage in the $Gd_{1.0}/Nd_x/TiO_2$ bimetallic NPs from x = 0.5 up to 6.0% has also led to higher $j_o$ values, which represents another line of evidence for improved catalytic performance for the HER. The kinetics of the HER became therefore faster if both Gd and Nd NPs were combined as bimetallic, $Gd/Nd/TiO_2$, rather than individually loaded on $TiO_2$, i.e., $Gd/TiO_2$ and $Nd/TiO_2$. Another supported bimetallic NPs electrocatalysts, such as Au–Pd [27,28], Au–Ni [29], Cu–Pt [30], and porous Cu–Ti [55] exhibited analogous findings for the HER.

Bimetallic catalysts exhibit eminent catalytic characteristics, which are not observed in their individual monometallic counterparts, through cooperative interactions (synergistic effects) [27–30,55]. Such catalytic characteristics comprise increased electrocatalytic activity, improved chemical/physical stability, a greater surface area, and increased catalyst selectivity. On these grounds, the HER catalytic activity of our synthesized $TiO_2$-supported bimetallic Gd/Nd catalyst is possibly a result of the synergistic effect between Gd and Nd, abundant catalytically active sites, and an increasingly accessible electrochemical surface area. The increased accessible electrochemical surface area of the investigated bimetallic catalysts, namely, $Gd_{1.0}/Nd_{0.5}/TiO_2$, $Gd_{1.0}/Nd_{1.0}/TiO_2$, $Gd_{1.0}/Nd_{3.0}/TiO_2$, and $Gd_{1.0}/Nd_{6.0}/TiO_2$ catalysts compared with their individual monometallic counterparts ($Gd/TiO_2$ and $Nd/TiO_2$) was evidenced from cyclic voltammetry measurements (Figure S3, Supporting Information).

As revealed from Table 10, the $Gd_{1.0}/Nd_{6.0}/TiO_2$ bimetallic nanocomposite, the best catalyst here, exhibited $E_{HER}$, $\eta_{10}$, and $j_o$ values of −22 mV vs. RHE, −109 mV dec$^{-1}$, and 0.72 mA cm$^{-2}$, respectively. These HER electrochemical kinetic parameter values are very close to those measured for the commercial Pt/C (−15 mV vs. RHE, −106 mV dec$^{-1}$, and 0.8 mA cm$^{-2}$). These findings reflect the outstanding HER catalytic performance of $Gd_{1.0}/Nd_{6.0}/TiO_2$ electrocatalyst that surpassed many effective electrocatalysts and are comparable with the most efficient ones reported in the literature (Table S1, Supporting Information).

An additional substantial electrochemical parameter, the Tafel slope, was also employed to assess and compare the HER catalytic efficiency of the investigated catalysts. In order to identify the major HER mechanism over the studied catalysts under such alkaline conditions, the measured Tafel slope values (Table 10) are contrasted with the standard values depicted in Equations (1)–(3). Such equations constitute the HER path in alkaline electrolytes on a particular catalyst [56]. The first water dissociation step (Volmer step, Equation (1)) is unavoidably the subject of the HER process under alkaline conditions, as few protons are present in alkaline electrolytes [56]. Following the Volmer step, either the two adsorbed hydrogen atoms are combined on the catalyst surface, forming a molecule $H_2$ (Tafel step, Equation (2)), or a hydrated proton is directly bonded to the adsorbed hydrogen

atom that requires the transfer of the electron from the catalyst surface (Heyrovsky step, Equation (3)).

$$H_2O + e^- \rightarrow H_{ads} + HO^- \qquad \text{(VolmerStep)}$$
$$b = \frac{2.34RT}{\alpha F} \cong 120 \text{ mV/dec} \tag{1}$$

$$H_{ads} + H_{ads} \rightarrow H_2 \qquad \text{(TafelStep)}$$
$$b = \frac{2.34RT}{(1+\alpha)F} \cong 30 \text{ mV/dec} \tag{2}$$

$$H_2O + e^- + H_{ads} \rightarrow H_2 + HO^- \qquad \text{(Heyrovsky)}$$
$$b = \frac{2.34RT}{2F} \cong 40 \text{ mV/dec} \tag{3}$$

Table 10 reports a Tafel slope value of 110 mV dec$^{-1}$ for the commercial Pt/C catalyst, which is in accordance with that recorded in the literature [57]; a good proof of the accuracy of the electrochemical measurements is utilized here. The Tafel lines of all tested Gd$_x$/TiO$_2$ and Nd$_x$/TiO$_2$ catalysts are parallel to each other with Tafel slopes of around 140 mV dec$^{-1}$. With Tafel slopes ranging from 108 to 113 mV dec$^{-1}$, the three studied Gd$_{1.0}$/Nd$_x$/TiO$_2$ bimetallic catalysts' Tafel lines are parallel to that of the Pt/C catalyst (110 mV dec$^{-1}$). The obvious decrease in the Tafel slope value from about 140 mV dec$^{-1}$ for Gd$_x$/TiO$_2$ and Nd$_x$/TiO$_2$ catalysts to about 110 mV dec$^{-1}$ for the four tested Gd$_{1.0}$/Nd$_x$/TiO$_2$ bimetallic catalysts adds another line of evidence for the enhanced HER kinetics on Gd$_{1.0}$/Nd$_x$/TiO$_2$ bimetallic catalyst surfaces. The explanation for this is that lower Tafel slopes usually indicate an abundance of active sites on the catalyst surface [58]. This result suggests an alkaline HER mechanism on the surface of the three studied bimetallic electrocatalysts that is similar to that taking place on the commercial Pt/C electrocatalyst. The Volmer step as the rate limiting step for the HER is part of that mechanism [57].

The catalysts' electrochemical active surface area (EASA) is another important metric used to compare their catalytic activity [59]. The inaccurate estimation of the specific capacitance of the composites, however, made it extremely difficult to quantify EASA for binary and ternary catalysts [60]. As a result of this, an alternative technique for assessing the catalytic activity of electrocatalysts is based on their electrochemical double-layer capacitance ($C_{dl}$), which has a direct link with EASA [61]. The values of $C_{dl}$ (Table 11) were calculated in this investigation using cyclic voltammetry (CV) measurements performed at various potential sweep rates, as mentioned in Section S1 (Supporting Information).

**Table 11.** Estimated values for the examined electrocatalysts' double-layer capacitance ($C_{dl}$), electrochemical active surface area (EASA), net voltammetry charge ($Q$), and number of active sites ($n$) based on CV measurements, Figure S3 (Supporting Information).

| Tested Cathode | $C_{dl}/$ $\mu F \text{ cm}^{-2}$ | EASA/cm$^2$ | $Q \times 10^3$/C | $n \times 10^8$/mol |
|---|---|---|---|---|
| TiO$_2$ NPs alone | 4.08 | 136.0 | 3.2 | 1.66 |
| Gd$_{0.5}$/TiO$_2$ | 8.26 | 275.3 | 7.3 | 3.78 |
| Gd$_{1.0}$/TiO$_2$ | 21.2 | 706.7 | 11.6 | 6.01 |
| Gd$_{3.0}$/TiO$_2$ | 28.8 | 960.0 | 25.2 | 13.06 |
| Gd$_{6.0}$/TiO$_2$ | 36.4 | 1213.3 | 34.5 | 17.88 |
| Nd$_{0.5}$/TiO$_2$ | 21.8 | 726.7 | 10.8 | 5.60 |
| Nd$_{1.0}$/TiO$_2$ | 29.9 | 996.7 | 15.4 | 7.98 |
| Nd$_{3.0}$/TiO$_2$ | 38.6 | 1286.7 | 31.7 | 16.43 |
| Nd$_{6.0}$/TiO$_2$ | 46.9 | 1563.3 | 41.2 | 21.35 |
| Gd$_{1.0}$/Nd$_{0.5}$/TiO$_2$ | 39.2 | 1306.7 | 22.9 | 11.87 |
| Gd$_{1.0}$/Nd$_{1.0}$/TiO$_2$ | 47.4 | 1580.0 | 43.5 | 22.54 |
| Gd$_{1.0}$/Nd$_{3.0}$/TiO$_2$ | 55.6 | 1853.3 | 48.6 | 25.19 |
| Gd$_{1.0}$/Nd$_{6.0}$/TiO$_2$ | 62.9 | 2096.7 | 56.8 | 29.43 |
| Pt/C | 65.4 | 2180.0 | 59.7 | 30.94 |

The results in Table 11 clearly showed that the $C_{dl}$ values rose as the tested NPs' doping % in $TiO_2$ increased, with Nd NPs being more efficient than Gd NPs at every doping percentage examined. Depending on the amount of Nd in the $Gd_{1.0}/Nd_x/TiO_2$, the $C_{dl}$ values further increased when Nd was co-doped with Gd. The obtained results demonstrated the catalytic influence of the synergistic interaction between Gd and Nd, as well as the abundance of catalytically active sites and a rising amount of accessible electrochemical surface area [27–30].

The surface sites that can be exploited for adsorption and desorption processes are more accessible and catalytically active in catalysts with higher $C_{dl}$ values [62].

The value of EASA was calculated from $C_{dl}$ using Equation (4) [62]:

$$\text{EASA} = C_{dl}/C_s \tag{4}$$

where $C_s$ is the specific capacitance for an electrode with 1.0 cm$^2$ of flat, uniform surface area; it is typically between 20 and 40 mF cm$^{-2}$. Table 11 summarizes the EASA values calculated for the materials under investigation using a flat electrode with an average value of 30 mF cm$^{-2}$.

At any measured RE doping %, the $Gd_{1.0}/Nd_x/TiO_2$ bimetallic NPs clearly achieved EASA values higher than those of their equivalent individuals, thus confirming the cooperative interactions (synergistic effects) between Gd and Nd in catalyzing the HER [27–30]. The highly active surface area of the $Gd_{1.0}/Nd_x/TiO_2$ catalysts (1307, 1580, 1853, and 2097 cm$^2$ for x = 0.5, 1.0, 3.0, and 6.0%, respectively) might have contributed to the appreciable rise in their $C_{dl}$ values (39.2, 47.4, 55.6, and 62.9 mF cm$^{-2}$ for x = 0.5, 1.0, 3.0, and 6.0%, respectively).

Cyclic voltammetry data, Figure S3, and Equation (5) [63], were employed to estimate the number of active sites *n* for the investigated materials.

$$n = Q/2F \tag{5}$$

where $F$ is the Faraday constant (96,485 C mol$^{-1}$) and 2 denotes the stoichiometric number of electrons that the HER of the electrode consumes. The studied $Gd_{1.0}/Nd_x/TiO_2$ bimetallic NPs (x = 0.5, 1.0, 3.0, and 6%) clearly displayed *n* values higher than those of their equivalent individual counterparts at any investigated RE doping percentage, namely, $Gd_x/TiO_2$ and $Nd_x/TiO_2$ (x = 0.5, 1.0, 3.0, and 6%). With increasing Nd content in the $Gd_{1.0}/Nd_x/TiO_2$ bimetallic NPs, the value of *n* increased approaching that computed for the commercial Pt/C electrocatalyst ($n = 30.94 \times 10^{-8}$ mol$^{-1}$) at a composition of Gd(1.0)-Nd (6.0), $n = 29.43 \times 10^{-8}$ mol$^{-1}$. These results confirmed the higher HER kinetics when the Gd and Nd NPs were combined into a hybrid bimetallic NP, $Gd_{1.0}/Nd_x/TiO_2$.

### 2.3.2. Faradaic Efficiency Calculations for the HER

The investigated catalysts' HER Faradaic efficiency (%) values were also calculated in order to further assess and compare their electrocatalytic activity. A controlled galvanostatic electrolysis was conducted to measure the amount of $H_2$ gas evolved ($V_m$, in mol) per hour using gas chromatography (GC), as reported in Section S4 (CGE), Equation (6).

$$V_m = \text{mol gas (GC)} \tag{6}$$

The value of $V_c$, the predicted amount of the released gas based on the charge transferred, is then computed using Equation (7) [64], assuming 100% Faradaic efficiency during the employed CGE:

$$V_c = Q(\text{CGE})/nF \tag{7}$$

where $F$ is the Faraday constant (96,485 C), $Q(\text{CGE})$ is a representation of the charge transferred through the WE during the CGE operation, and $n$ ($2H^+ + 2e = H_2$, $n = 2$) is a mathematical representation of the number of electrons exchanged during the HER. The value of

$\varepsilon$ is derived by dividing $V_m$ by $V_c$. The measured electrocatalyst's Faradaic efficiency ($\varepsilon\%$) is then calculated by multiplying the ratio ($V_m/V_c$) quotient by 100, Equation (8) [64].

$$\text{Faradaic efficiency } (\varepsilon\%) = [Fn \text{ (mol gas(GC))}100]/Q(\text{CGE}) \tag{8}$$

The calculated and measured quantities of $H_2$ evolved for the examined electrocatalysts during the first hour of CGE are summarized in Table 12, which revealed that the tested $Gd_{1.0}/Nd_x/TiO_2$ bimetallic NPs, x = 1.0 and 6%, exhibited $\varepsilon\%$ values that are much higher than those of their equivalent individuals at any studied RE doping percentage, $Gd_x/TiO_2$ and $Nd_x/TiO_2$ (x = 1.0 and 6%). Additionally, it was also noticed that as Nd content was increased in $Gd_{1.0}/Nd_x/TiO_2$ from 1.0 to 6.0%, the HER's $\varepsilon\%$ value also enhanced from 92.9 to 98.7%, thus approaching that of the commercial Pt/C electrocatalyst (99.5%). These results provided another piece of evidence for the enhanced HER kinetics when both Gd and Nd dopants were brought together on $TiO_2$, forming the $Gd_{1.0}/Nd_x/TiO_2$ bimetallic NPs rather than being loaded separately, i.e., $Gd_{1.0}/TiO_2$ and $Nd_x/TiO_2$.

**Table 12.** Mean value (standard deviation) of $V_{H2}$ (measured and calculated) obtained after 1 h of a controlled galvanostatic electrolysis (CGE) *, together with the Faradaic efficiency values, *FE* (%), for the studied catalysts.

| Tested Catalyst | H$_2$ Measured by GC (H$_2$/μmol h$^{-1}$) | Calculated H$_2$ Based on the Charge Passed during Electrolysis | | *FE* (%) |
|---|---|---|---|---|
| | | Charge Passed/C | H$_2$/μmol h$^{-1}$ | |
| TiO$_2$ NPs alone | 6.9 (0.12) | 2.4 (0.05) | 12.4 (0.2) | 55.4 (0.8) |
| Gd$_{1.0}$/TiO$_2$ | 11.8 (0.15) | 3.1 (0.055) | 15.9 (0.31) | 74.2 (1.1) |
| Gd$_{6.0}$/TiO$_2$ | 15.6 (0.3) | 3.6 (0.06) | 18.7 (0.38) | 83.6 (1.3) |
| Nd$_{1.0}$/TiO$_2$ | 14.3 (0.26) | 3.4 (0.052) | 17.8 (0.35) | 80.5 (1.2) |
| Nd$_{6.0}$/TiO$_2$ | 20.1 (0.35) | 4.3 (0.07) | 22.4 (0.4) | 89.8 (1.5) |
| Gd$_{1.0}$/Nd$_{1.0}$/TiO$_2$ | 22.9 (0.4) | 4.8 (0.09) | 24.6 (0.42) | 92.9 (1.4) |
| Gd$_{1.0}$/Nd$_{6.0}$/TiO$_2$ | 31.4 (0.55) | 6.1 (0.12) | 31.8 (0.5) | 98.7 (1.6) |
| Pt/C | 32.9 (0.3) | 6.4 (0.1) | 33.1 (0.36) | 99.5 (1.4) |

* CGE: the catalyst is held at a current density of $-10$ mA cm$^{-2}$ for 1 h in 1.0 M KOH solution at 25 °C.

### 2.4. Best Catalyst's Long-Term Stability Tests

Excellent electrocatalysts must meet a number of criteria, one of which is long-term stability. To assess the stability of the best catalyst for the HER, two main electrochemical approaches were used. They comprise 72 h of controlled potential electrolysis (chronoamperometry) measurements as well as continuous (repetitive) cyclic polarization (CP) up to 10,000 cycles, Figure 14.

It follows from the CP measurements, Figure 14, that the catalyst's polarization curve maintained a high degree of similarity with just minor current losses after 10,000 cycles, thusreflecting good stability in the catalytic activity. CP findings were validated by electrolysis results at a static overpotential (inset of Figure 14); the current remained essentially constant during the run.

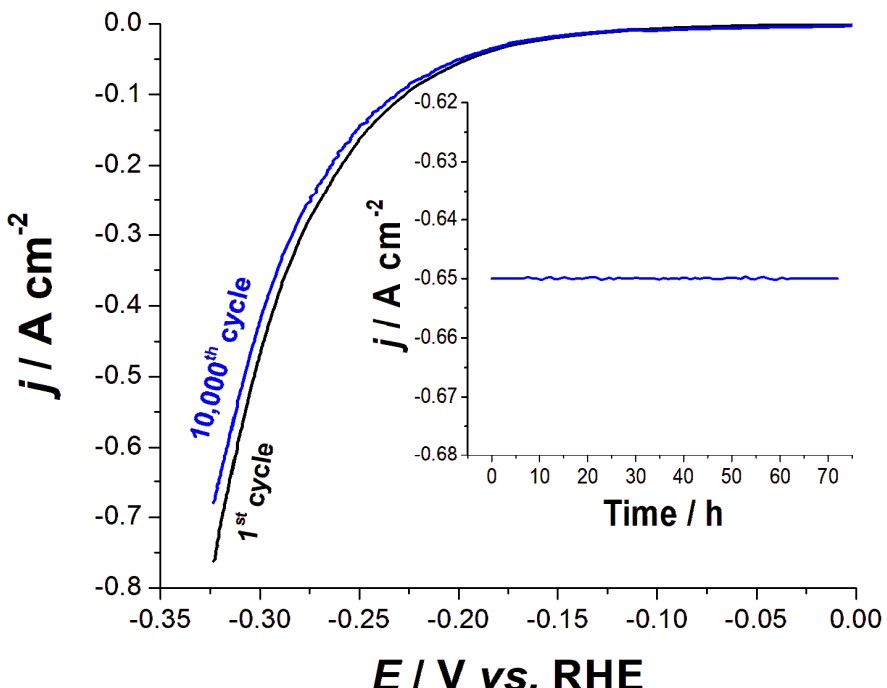

**Figure 14.** Long-term stability tests recorded for the best performing electrocatalyst ($Gd_{1.0}/Nd_{6.0}/TiO_2$) in 1.0 M KOH solution at room temperature for the HER. LSV measurements were conducted at a scan rate of 50 mV s$^{-1}$. Insets are chronoamperometry measurements (*j* vs. *t*) performed on the catalyst at a constant applied potential of −0.32 V vs. RHE.

## 3. Experimental

### 3.1. Synthesis of $Gd^{3+}$- or $Nd^{3+}$- as Well as 0.5%$Gd^{3+}$-0.5%$Nd^{3+}$-Doped $TiO_2$ NPs

Nanocrystalline $TiO_2$ nanoparticles (NPs) doped with different atomic concentrations (0.5, 1.0, 3.0, and 6.0%) of $Gd^{3+}$ and $Nd^{3+}$ ions were prepared adopting our previous method for the synthesis of pure $TiO_2$ nanoparticles [51]. The obtained materials were referred to as x% Gd or x% Nd-doped $TiO_2$, where %x is the percent content of $Gd^{3+}$ or $Nd^{3+}$ ions in the sample. Mixed 0.5%$Gd^{3+}$-0.5%$Nd^{3+}$-doped $TiO_2$ NPs were also obtained by the same method using a mass ratio of 0.5% of both $Gd^{3+}$ and $Nd^{3+}$ ions.

### 3.2. Electrocatalytic Activity Measurements
Electrochemical Setup

Electrochemical characterizations were carried out using a standard double-jacketed three-electrode electrochemical cell. A graphite rod (99.999% pure, Sigma-Aldrich, Darmstadt, Germany) and mercury/mercury oxide, Hg/HgO, NaOH(0.1 M), served as the cell's auxiliary and reference electrodes, respectively. The working electrode was a 3 mm glassy carbon (GC) loaded with catalyst (WE). Section S2 of the Supporting Information file gives the full description of the WE preparation for electrochemical experiments.

To evaluate the performance and stability of the synthesized electrocatalysts toward the HER, various electrochemical techniques were used, as reported in Section S3 (Supporting Information).

The electrochemically active surface area (EASA) of the catalysts was estimated using cyclic voltammetry (CV) tests carried out at various potential scan rates (ν: 20–120 mV s$^{-1}$) covering the potential range (0.32–0.42 V vs. RHE), which only permits the capacitive current to flow. The catalyst $C_{dl}$ can be determined by plotting the difference in current density between anodic and cathodic scans ($\Delta J = J_{anodic} - J_{cathodic}$) against the slope of the $\Delta J$ vs. ν plot at 0.37 V vs. RHE.

## 4. Conclusions

In this work, a facile and efficient one-pot method for the synthesis of $RE^{3+}$ ($Gd^{3+}$ or $Nd^{3+}$)-doped $TiO_2$ NPs with different atomic concentrations (ca. 0.5–6.0%), namely, $Gd_x/TiO_2$, $Nd_x/TiO_2$, and $Gd_{1.0}/Nd_x/TiO_2$ bimetallic NPs, x = 0.5, 1.0, 3.0, and 6%, as efficient cathode materials for $H_2$ production, was developed. The structure and morphology of the obtained materials were characterized by using various techniques, which indicated that the prepared RE-doped $TiO_2$ NPs were pure-phase and uniformly dispersed. The crystallite size was about 20 nm in the case of $Gd^{3+}$-doping, while doping with $Nd^{3+}$ decreased the crystallite size to about 10 nm. Linear cathodic polarization measurements were used to examine the as-prepared NPs as active electrocatalysts for effective hydrogen generation in alkaline solution (1.0 M KOH). The studied $TiO_2$-doped bimetallic NPs, namely, $Gd_{1.0}/Nd_x/TiO_2$, x = 0.5, 1.0, 3.0, and 6%, showed higher HER catalytic performance than their corresponding individual counterparts at any tested RE doping percentage, namely, $Gd_x/TiO_2$ and $Nd_x/TiO_2$. The $Gd_{1.0}/Nd_x/TiO_2$ HER catalytic performance was enhanced with increasing Nd content from 0.5 up to 6.0%. The $Gd_{1.0}/Nd_x/TiO_2$ maximum HER catalytic activity was attained at x = 6% with HER electrochemical kinetic parameters (onset potential: $-22$ mV, Tafel slope: 109 mV dec$^{-1}$, and exchange current density: 0.72 mA cm$^{-2}$), approaching the performance of the commercial Pt/C electrocatalyst (onset potential: $-15$ mV, Tafel slope: 106 mV dec$^{-1}$, and exchange current density: 0.80 mA cm$^{-2}$). The synergistic interaction of Gd and Nd is thought to be the major cause of the bimetallic catalyst's activity. The simplicity and originality of the procedure adopted here to synthesize such hybrid NPs, together with their high HER catalytic activity in the dark, are the fundamental features of this work.

**Supplementary Materials:** The following supporting information can be downloaded at: https://www.mdpi.com/article/10.3390/catal13081192/s1, Figure S1. FT-IR spectra of DMSO, Gd3+.nBuO, and Gd-doped $TiO_2$ NPs before (red) and after (blue) annealing process; Figure S2. TGA-DTA curves of Gdx-doped $TiO_2$ NPs recorded from room temperature to 1000 °C before (a) and after (b) calcination; Figure S3. Cyclic voltammograms recorded for studied catalysts at various potential scan rates (20–100 mV s$^{-1}$) measured in a non-Faradaic region of the voltammograms. Measurements were conducted in deaerated KOH solution (1.0 M) at room temperature; Figure S4. Double-layer capacitance measurements for determining the electrochemically-active surface area of Gdx/TiO$_2$ N and Ndx/TiO$_2$ N (x = 0.5, 1.0, 3.0, and 6.0 %) catalysts; Figure S5. Double-layer capacitance measurements for determining the electrochemically-active surface area of Gd1.0/Ndx/TiO$_2$ N (x = 0.5, 1.0, 3.0, and 6.0 %) catalysts; Table S1: Comparison of HER catalytic activity of our best performing electrocatalysts with the highly efficient ones reported in the literature in alkaline solutions [65–87].

**Author Contributions:** Conceptualization, M.A., N.A.A. and H.M.O.A.-M.; methodology, S.I.A., M.R.D. and A.A.S.; software (XRD, XRF, and XPS) and formal analysis, H.M.O.A.-M.; investigation, M.A. and N.A.A.; resources, M.A.A. and M.M.I.; data curation, M.A., R.B., M.A.A. and M.M.I.; writing—original draft preparation, A.A.S., M.A.A. and M.M.I.; writing—review and editing, M.A.A. and M.M.I.; visualization, M.A.A. and M.M.I.; supervision, M.A., M.A.A. and M.M.I.; project administration, M.A.; funding acquisition. All authors have read and agreed to the published version of the manuscript.

**Funding:** The researchers would like to acknowledge Deanship of Scientific Research, Taif University for funding the work.

**Conflicts of Interest:** The authors declare no conflict of interest.

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
