# Peer review of "Enhanced Alkaline Hydrogen Evolution on Gd1.0/Ndx (x = 0.5, 1.0, 3.0, and 6.0%)-Doped TiO2 Bimetallic Electrocatalysts"

_catalysts, doi:10.3390/catal13081192_

Round 1

Reviewer 1 Report

In this manuscript, the author developed RE3+ (Gd3+ or Nd3+)-doped TiO2 NPs with different atomic concentrations as efficient cathode materials for H2 production and emphasized that the synergistic interaction of Gd and Nd is thought to be the major cause of the bimetallic catalyst's activity. However, physical insights need to be clarified and more supports should be added to strengthen the argument. Major revisions are needed before potential consideration of future acceptance. The specific comments are listed as follows:

(1) In the XRD Rietveld refinements, the volume of the anatase tetragonal cell featured a slight increase with increasing the Gd and Nd content, why does the crystallite size decrease in contrast?

(2) As we know, the radius of Nd is larger than Gd, so why do the cell parameters and crystallite size of Gdx/TiO2 is larger than those of Ndx/TiO2 (Table 3)?

(3) In the XPS analysis, why does the decreased intensity of the O 1s peak after doping with Gd or Nd compared with the pure TiO2 can confirm the successful doping in TiO2?

(4) In the XRD analysis, the author claimed that the doping of Gd and Nd replaces the Ti atom inside the anatase lattice, indicating that the Gd or Nd locates at the octahedral interstice of O species. Why could the shifts in the BE of Ti 2p and O 1s peaks be ascribed to the formation of Ti-Gd and Ti-Nd bonds on the grain boundaries of the crystallites? Please give more evidence to support the presence of Ti-Gd and Ti-Nd bonds on the grain boundaries.

(5) In Table 4-7, why do the atomic ratios of O:Ti are approximately 3:1 in the XPS results, far from the stoichiometric ratio of TiO2?

(6) From the SEM images, how does the author estimate the average size of Gdx/TiO2 and Ndx/TiO2 samples is about 14.0 nm based on the low-magnified images with no more than 2500X?

(7) From the EDX results in Table 8, why does the atomic ratio of O:Ti lower than 1:1 according to the mass ratio of O and Ti, which is fully contradictory to the XPS results?

(8) In Figure 10, it is obvious that the average particle size of 0.5%Gd-TiO2 is more than 100 nm instead of 10-12 nm, which is very interesting and needs further investigation. Besides, it would be better to use the TEM images with the same magnification for these samples.

(9) In the characterization section, the author analyzed some physicochemical properties of Gdx/TiO2, Ndx/TiO2, and Gd0.5/Nd0.5/TiO2 NPs. However, in the electrochemical discussion, the best HER performance was obtained from the Gd1.0/Ndx/TiO2, why does the corresponding properties of these samples not be discussed in the characterization section?

(10) The abbreviation of the electrochemical active surface area consists of ECSA and EASA in this manuscript, which should be unified.

(11) In Table S1, all the listed references were reported before 2020, it would be better to compare the HER performance with currently reported catalysts.

(12) In Table 11, the ECSA value should be lower than 3 cm2 if the Cs value was set as 30 mF cm-2 in this manuscript, which is a mistake.

(13) On line 488, ‘Table S12’ should be ‘Table 12’.

(14) In the stability test, why does the current density just 0.65 A cm-2 at a constant applied potential of −0.5 V vs. RHE, which is far inconsistent with the LSV curves?

(15) In the supporting information, there is only 21 cited references, but Table S1 has reference [22] and [23].

(16) Many format errors in the manuscript.

(17) Similar works on the same topic should be discussed, e.g.,            https://doi.org/10.1039/D0CS00575D, doi.org/10.1038/s41929-021-00578-1,  https://doi.org/10.1021/acsaem.8b01365

The writing should be improved 

Author Response

Responses to the comments raised by Reviewer # 1

In this manuscript, the author developed RE3+ (Gd3+ or Nd3+)-doped TiO2 NPs with different atomic concentrations as efficient cathode materials for H2 production and emphasized that the synergistic interaction of Gd and Nd is thought to be the major cause of the bimetallic catalyst's activity. However, physical insights need to be clarified and more supports should be added to strengthen the argument. Major revisions are needed before potential consideration of future acceptance. The specific comments are listed as follows:

(1) In the XRD Rietveld refinements, the volume of the anatase tetragonal cell featured a slight increase with increasing the Gd and Nd content, why does the crystallite size decrease in contrast?

The data presented in Table 3 does not show any correlation between the crystallite size to the increase of Gd and Nd content. In the case of doping with Gd, the crystallite size average was about 20 nm while in the case of Nd, the average was about 10 nm. This can be attributed to the different effects of doping with Gd and Nd on the TiO2 nanoparticles. Also, as far as we know, there is not a clear relation between the size of crystallites in a nanomaterial and the size of the unit cell of its lattice. Fundamentally, the size of the crystallite is determined by the preparation conditions (preparation route, temperature, …), while the lattice parameters are determined by atomic/ionic radii, symmetry aspects of the lattice, etc.

(2) As we know, the radius of Nd is larger than Gd, so why do the cell parameters and crystallite size of Gdx/TiO2 is larger than those of Ndx/TiO2 (Table 3)?

We agree with this valuable observation of the reviewer. Actually, Nd3+ is larger than Gd3+ by about 0.04 Å which is about 4% only. On the other hand, the average cell of Nd/TiO2 is smaller than that of Gd/TiO2 by about 1 Å3 which is about 0.7% of the TiO2 cell volume. Considering that Nd-doping has reduced the crystallite size of TiO2 to about 10 nm i.e., 50% of the crystallite size in the case of Gd-doping and this is manifested in the noticeable broadening of the diffraction peaks in Nd/TiO2 in comparison to Gd/TiO2. This larger broadening affects the diffraction peak position and in turn the extracted cell parameters and volume. We believe that a 0.7% difference can be accepted for the current case, and we can consider that both Gd and Nd increase the cell volume of TiO2 in almost the same way. Especially, since the volume of the cell is increasing as the doping level increases in each case of doping with Nd and Gd.

(3) In the XPS analysis, why does the decreased intensity of the O 1s peak after doping with Gd or Nd compared with the pure TiO2 can confirm the successful doping in TiO2?

The intensity of the peaks mainly changes because of two reasons: (1)  the intensity is directly linked to the number of atoms with the respective oxidation state and (II) directly proportional to atoms on the respective chemical state. In our study the after doping of the Gd or Nd the chemical environment has been changed. The decrease of the intensity of the O 1s peak may be due to the M-O interaction.

(4) In the XRD analysis, the author claimed that the doping of Gd and Nd replaces the Ti atom inside the anatase lattice, indicating that the Gd or Nd locates at the octahedral interstice of O species. Why could the shifts in the BE of Ti 2p and O 1s peaks be ascribed to the formation of Ti-Gd and Ti-Nd bonds on the grain boundaries of the crystallites? Please give more evidence to support the presence of Ti-Gd and Ti-Nd bonds on the grain boundaries.

  • In the case of high contents of the doped Gd and Nd, the shifts in the BE of the Ti 2p to high binding energy with Gd and Nd could be attributed to the either new species of TiO2, or the inelastic scatting of Gd and Nd within the anatase crystal structure.
  • The shifts of the O 1s could be attributed to increase of the concentrations of Gd and Nd from 0.5% to 6% in the anatase phase, which indicates the presence of Gd2O3 and Nd2O3 in the anatase crystal. Additionally, the remarkable shifts of the O1s peak could be attributed to the surface relaxation effects.

This was illustrated in section 3.1. Characterization of Gdx/TiO2, Ndx/TiO2, and Gd0.5/Nd0.5/TiO2 NPs

(5) In Table 4-7, why do the atomic ratios of O:Ti are approximately 3:1 in the XPS results, far from the stoichiometric ratio of TiO2?

  • As well known that, the XPS is only valid for the surface analysis of a thickness of few nm (1-10 nm) which could be completely different form the bulk material. The carbon contamination is a well-known effect during XPS measurement and often plays a role in surface chemistry. During the measurement of the XPS analysis carbon contamination is there and oxygen is also associate with the carbon. The oxygen associated with the surface contaminated carbon is also contributed on the atomic percentage of the oxygen. (Journal of Vacuum Science & Technology A 39, 043203 (2021) https://doi.org/10.1116/6.0001013)
  • Due to the chemical treatment of the different concentration of TiO2 as well as the Gd3+ and Nd3+ in the TiO2 nanoparticles mentioned in our previous work [Ref. 34], the final atomic ratios of O:T is relatively nonstoichiometric.
  • This was illustrated in the manuscript, page 11.

[34] Ibrahim, M.M.; Mezni, A.; Alsawat, M.; Kumeria, T.; Das, M.R.; Alzahly, S.; Aldalbahi, A.; Gornicka, K.; Ryl, J.; Amin, M.A.; T. Altalhi. Enhanced hydrogen evolution reaction on highly stable titania‐supported PdO and Eu2O3 nanocomposites in a strong alkaline solution, Int. J. Energy Res., 2019, 43, 5367-5383.

(6) From the SEM images, how does the author estimate the average size of Gdx/TiO2 and Ndx/TiO2 samples is about 14.0 nm based on the low-magnified images with no more than 2500X?

Due to the typing error and because of the TEM images not SEM images, the sentence “According to the statistical estimation, the average size was about 14.0 nm, which was in accordance with the value determined by XRD (18.94 nm).” has been omitted.

(7) From the EDX results in Table 8, why does the atomic ratio of O:Ti lower than 1:1 according to the mass ratio of O and Ti, which is fully contradictory to the XPS results?

The SEM-EDX of 0.5%Gd-TiO2 and 0.5%Nd-TiO2 has been re-measured again (Please see Table 8 after modification).

(8) In Figure 10, it is obvious that the average particle size of 0.5%Gd-TiO2 is more than 100 nm instead of 10-12 nm, which is very interesting and needs further investigation. Besides, it would be better to use the TEM images with the same magnification for these samples.

The TEM image of 0.5%Gd-TiO2 (Figure 10) has been improved and modified (Please see F after modification).

(9) In the characterization section, the author analyzed some physicochemical properties of Gdx/TiO2, Ndx/TiO2, and Gd0.5/Nd0.5/TiO2 NPs. However, in the electrochemical discussion, the best HER performance was obtained from the Gd1.0/Ndx/TiO2, why does the corresponding properties of these samples not be discussed in the characterization section?

(10) The abbreviation of the electrochemical active surface area consists of ECSA and EASA in this manuscript, which should be unified.

We considered EASA as the abbreviation of the electrochemical active surface area throughout the manuscript.

(11) In Table S1, all the listed references were reported before 2020, it would be better to compare the HER performance with currently reported catalysts.

We compared the HER performance with currently and recently reported catalysts, 2021, 2022, and 2023, as shown in references [22-25] in the SI file.

(12) In Table 11, the ECSA value should be lower than 3 cm2 if the Cs value was set as 30 mF cm-2 in this manuscript, which is a mistake.

Yes, that’s right. We found a mistake in Table 11. Thanks a lot for this remark. Indeed, the unit of Cdl in Table 11 is actually expressed in µF cm-2, and not mF cm-2. We corrected that mistake via considering µF cm-2 instead of mF cm-2. Based on this, the Cs value was set as 0.03 µF cm-2 which is equivalent to 30 mF cm-2. For example, for Cdl = 4.08 µF cm-2, EASA = (4.08 µF cm-2) / (0.03 µF cm-2) = 136 cm2.   

(13) On line 488, ‘Table S12’ should be ‘Table 12’.

We corrected that.

(14) In the stability test, why does the current density just 0.65 A cm-2 at a constant applied potential of −0.5 V vs. RHE, which is far inconsistent with the LSV curves?

The current density recorded a value of 0.65 A cm-2 at a constant applied potential of −0.32 V vs. RHE , and not @ -0.5 V vs. RHE. We corrected that in the text.

(15) In the supporting information, there is only 21 cited references, but Table S1 has reference [22] and [23].

References [22] and [23] mentioned in Table S1 actually refer to the same work in Ref. [21], Liu et al., A highly efficient alkaline HER Co–Mo bimetallic carbide catalyst with an optimized Mo d-orbital electronic state. J. Mater. Chem. A, 2019, 7, 12434-12439, and we replaced references [22] and [23] by [21].

(16) Many format errors in the manuscript.

We addressed such format errors in the manuscript.

(17) Similar works on the same topic should be discussed, e.g.,            https://doi.org/10.1039/D0CS00575D, doi.org/10.1038/s41929-021-00578-1,  https://doi.org/10.1021/acsaem.8b01365

We cited this work. See references [15-17]:

[15] Wang, J.; Gao, Y.; Kong, H.; Kim, J.; Choi, S.; Ciucci, F.; Hao, Y.; Yang, S.; Shao, Z.; Lim, J. Non-precious-metal catalysts for alkaline water electrolysis: operando characterizations, theoretical calculations, and recent advances. Chem. Soc. Rev., 2020,49, 9154-9196.

[16] Wang, J.; Kim, S.-J.; Liu, J.; Gao, Y.; Choi, S.; Han, J.; Shin, H.; Jo, S.; Kim, J.; Ciucci, F.; Kim, H.; Li, Q.; Yang, W.; Long, X.; Yang, S.; Cho, S.-P.; Chae, K. H.; Kim, M. G.; Kim, H.; Lim, J.; Redirecting dynamic surface restructuring of a layered transition metal oxide catalyst for superior water oxidation, Nature Catalysis, 2021, 4, 212–222.

[17] Wang, J.; Gao, Y.; Ciucci, F. Mechanochemical Coupling of MoS2 and Perovskites for Hydrogen Generation ACS Appl. Energy Mater. 2018, 1, 11, 6409–6416.

Reviewer 2 Report

The manuscript reported by Alsawat, Mohammed et al provides a series of Nd and Gd doped TiO2 for hydrogen evolution reaction (HER). Their sample shows excellent HER performance.This reviewer recommends this manuscript should be suitable for publication in catalysts after a major revision.

Comments:

1.In Fig. 4b, the XRD pattern of 3% Nd/TiO2 is noisier than others. Why? 

2. Why we could find a significant energy shift in Figure 6BCD between 0.5% Gd and 6.0% Gd samples on the same species in Ti, O and Gd spectra?

3.Why the HER overpotential at j = 10 mA cm-2 of Pt/C in KOH is 106 mV which is higher than other references?

4.Why you have two figure 12 in the work?

5.In your stability test, 70 s is too short. Authors should make a longer test at the constant applied potential (at least over 20 hours).

6.Authors should provide EXAFS to study the local bonding environments of Nd and Gd species.

7.The manuscript writing could be further polished.

Normal level, should be further polished.

Author Response

Responses to the comments raised by Reviewer # 2

The manuscript reported by Alsawat, Mohammed et al provides a series of Nd and Gd doped TiO2 for hydrogen evolution reaction (HER). Their sample shows excellent HER performance.This reviewer recommends this manuscript should be suitable for publication in catalysts after a major revision.

Comments:

1.In Fig. 4b, the XRD pattern of 3% Nd/TiO2 is noisier than others. Why?

This may be attributed to the smaller amount of the powder used in the XRD measurement of this sample. 

  1. Why we could find a significant energy shift in Figure 6BCD between 0.5% Gd and 6.0% Gd samples on the same species in Ti, O and Gd spectra?
  • In the case of high contents of the doped Gd and Ni, the shifts in the BE of the Ti 2p to high binding energy with Gd and Nd could be attributed to the either new species of TiO2, or the inelastic scatting of Gd and Nd with anatase crystal structure.
  • The shifts of the O 1s could be attributed to increase of the concentrations of Gd and Nd from 0.5% to 6% in the anatase phase, which indicates to the presence of Gd2O3 and Nd2O3 in the anatase crystal. Additionally, the remarkable shifts of the O1s peak could be attributed to the surface relaxation effects.

This was illustrated in section 3.1. Characterization of Gdx/TiO2, Ndx/TiO2, and Gd0.5/Nd0.5/TiO2 NPs

3.Why the HER overpotential at j = 10 mA cm-2 of Pt/C in KOH is 106 mV which is higher than other references?

This value (106 mV @ 10 mA cm-2) was the average value obtained after repeating the run at least three times. In addition, this value is close to that is previously reported in the literature, 111 mV @ 10 mA cm-2, (Lu, Q., Hutchings, G., Yu, W. et al. Highly porous non-precious bimetallic electrocatalysts for efficient hydrogen evolution. Nat Commun 6, 6567 (2015). https://doi.org/10.1038/ncomms7567). Indeed, the reproducibility of the electrochemical measurements was verified by performing each run at least three times, during which statistically significant results were collected. Each reported value's arithmetic mean and standard deviation were computed.

4.Why you have two figures 12 in the work?

 We corrected that mistake.

5.In your stability test, 70 s is too short. Authors should make a longer test at the constant applied potential (at least over 20 hours).

Our stability test duration is actually 3 days (72 hours), and not 70 s. It was a typing mistake in the x-axis of the figure that is located in the inset of Fig. 14, and we corrected that mistake.

6.Authors should provide EXAFS to study the local bonding environments of Nd and Gd species.

As the extended X-ray Absorption fine structure (EXAFS) need a beam time at the synchrotron light sources, which need further time. Therefore, we decided to use the X-ray photoelectron spectroscopy (XPS) as an alternative technique to demonstrate the elemental speciation in the present samples. In our future work, we are planning to use EXAFS once we received a beamtime.

7.The manuscript writing could be further polished.

We polished the manuscript’s language.

Round 2

Reviewer 1 Report

the authors have fully addressed my concerns and it can be accepted now 

Reviewer 2 Report

accepted

normal level